# Isometric Quotient Variational Auto-Encoders for Structure-Preserving Representation Learning

**In Huh**[1,*]**, Changwook Jeong**[2,†]**, Jae Myung Choe**[1]**, Young-Gu Kim**[1]**, Dae Sin Kim**[1]
[1]CSE Team, Innovation Center, Samsung Electronics
[2]Graduate School of Semiconductor Materials and Devices Engineering, UNIST
[*]in.huh@samsung.com [†]changwook.jeong@unist.ac.kr

## Abstract

We study structure-preserving low-dimensional representation of a data manifold embedded in a high-dimensional observation space based on variational auto-encoders (VAEs). We approach this by decomposing the data manifold $\mathcal{M}$ as $\mathcal{M} = \mathcal{M}/G \times G$, where $G$ and $\mathcal{M}/G$ are a group of symmetry transformations and a quotient space of $\mathcal{M}$ up to $G$, respectively. From this perspective, we define the structure-preserving representation of such a manifold as a latent space $\mathcal{Z}$ which is isometrically isomorphic (i.e., distance-preserving) to the quotient space $\mathcal{M}/G$ rather $\mathcal{M}$ (i.e., symmetry-preserving). To this end, we propose a novel auto-encoding framework, named *isometric quotient VAEs (IQVAEs)*, that can extract the quotient space from observations and learn the Riemannian isometry of the extracted quotient in an unsupervised manner. Empirical proof-of-concept experiments reveal that the proposed method can find a meaningful representation of the learned data and outperform other competitors for downstream tasks.

## 1   Introduction

There has been a common consensus that natural image datasets form a low-dimensional manifold $\mathcal{M}$ embedded in a high-dimensional observation space $\mathbb{R}^d$, i.e., $\dim(\mathcal{M}) \ll d$ [6]. From this perspective, a good neural network is a mapping function that can recognize the underlying structure of the data manifold well [25]. A question that arises is what structures should be represented from the data.

For unsupervised learning task, especially in generative modeling, if we suppose the data manifold to be a Riemannian one, then preserving geometric structures in the representation space is a key consideration [1, 8, 25]. To make things more explicit, if $\mathcal{Z}$ is a latent representation of $\mathcal{M}$, then $\mathcal{Z}$ should be isometrically isomorphic to $\mathcal{M}$ in the sense of the Riemannian isometry. This means that an infinitesimal distance on the data manifold should be preserved in the latent space as well, promoting geodesic distances in the latent space that accurately reflect those in the data space. Recently, several papers [8, 40, 25] have suggested using the concept of Riemannian isometry for deep generative models, such as variational auto-encoders (VAEs) [22] and generative adversarial networks (GANs) [14], to obtain more meaningful and relevant representations. These concepts have been applied to improve latent space interpolations [30] or clustering [45] of data manifolds.

However, the Riemannian isometry does not tell the whole story of the data structure, particularly in the case of vision datasets. One common property of natural images is that there are symmetry transformations that preserve the semantic meaning of the given image. For example, if one applies a rotational transformation to a certain image, then its semantic meaning, e.g., its label, does not change. In this situation, what one really wants to do is represent the inherent geometry of the manifold, i.e., the geometry up to such symmetry transformations [29].

37th Conference on Neural Information Processing Systems (NeurIPS 2023).

The concept of the inherent geometry can be formalized by using the notion of the principal bundle $\mathcal{M} = \mathcal{M}/G \times G$, a fiber bundle that consists of the group $G$ of symmetry transformations and a quotient space $\mathcal{M}/G$ as the fiber and base spaces, respectively [16, 26]. In this case, the inherent geometry of $\mathcal{M}$ up to $G$ is indeed determined by its quotient $\mathcal{M}/G$ solely; the quotient formulates an invariant structure of the data, thus a measure on $\mathcal{M}/G$ gives a measure on $\mathcal{M}$ that is invariant to the actions of $G$. Therefore, one should find the Riemannian isometry of $\mathcal{M}/G$ rather than $\mathcal{M}$ for a geometric-meaningful representation of the dataset (see Figure 1). Nevertheless, to the best of our knowledge, there has been a lack of studies, at least in the field of generative models, that integrate both the concepts of the quotient space and its Riemannian geometry.

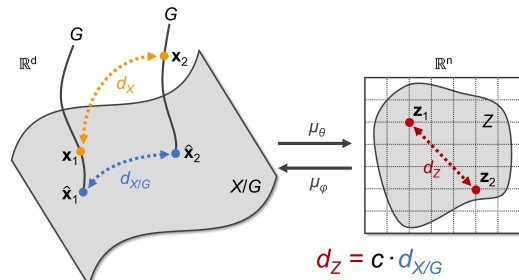

Figure 1: We denote $\mathcal{X} \subset \mathbb{R}^d$ as an extrinsic view of the data manifold $\mathcal{M}$. It consists as $\mathcal{X} = \mathcal{X}/G \times G$ where $\mathcal{X}/G$ is a quotient space of $\mathcal{X}$ and $G$ is a group of symmetry transformations that naturally act on $\mathcal{X}$, respectively. From this perspective, the structure-preserving representation $\mathcal{Z}$ should preserve all geodesic distances of $\mathcal{X}/G$ ($d_{\mathcal{X}/G}(\cdot, \cdot)$) rather than them of $\mathcal{X}$ ($d_{\mathcal{X}}(\cdot, \cdot)$), i.e., $d_{\mathcal{Z}}(\mathbf{z}_1, \mathbf{z}_2) = c \cdot d_{\mathcal{X}/G}(\hat{\mathbf{x}}_1, \hat{\mathbf{x}}_2)$, $\forall \hat{\mathbf{x}}_1, \hat{\mathbf{x}}_2 \in \mathcal{X}/G, \mathbf{z}_1, \mathbf{z}_2 \in \mathcal{Z}$, where $d_{\mathcal{Z}}(\cdot, \cdot)$ is a distance on $\mathcal{Z}$. Such a space $\mathcal{Z}$ preserves all the essential geometry and symmetry of the data.

In this paper, we propose a novel auto-encoding framework, named *isometric quotient VAEs (IQ-VAEs)*, that can extract the quotient space from observations and learn its Riemannian isometry in an unsupervised manner, allowing for a correct representation of the data manifold and accurate measurement of distances between samples from the learned representation. The proposed IQVAE model can be applied to a variety of practical tasks such as downstream clustering and out-of-distribution (OoD) detection. We evaluate the model's performance using three datasets: rotated MNIST [44], mixed-type wafer defect maps (MixedWM38) [42] and cervical cancer cell images (SIPaKMeD) [32]. In summary, our contribution is threefold:

- We propose a novel approach for structure-preserving representation learning by viewing a data manifold as a principal bundle, and formulate it as an unsupervised learning task that finds a Riemannian isometry for the quotient of the data manifold.

- We introduce a practical method, called the IQVAE, which can learn such a structure-preserving representation in an unsupervised manner by using the auto-encoding framework.

- We demonstrate through experimental evaluations on various datasets that our proposed method can find useful representations and outperform other competing methods for downstream tasks such as classification, clustering, and OoD detection.

## 2 Related Works

### 2.1 $G$-Invariance and Quotient Space Learning

Quotient space learning is highly related to the notion of the $G$-invariance. Data augmentation is the most common approach to deal with the $G$-invariance of label distributions [18]. To achieve such a symmetry more explicitly, various $G$-invariant and $G$-equivariant architectures have been proposed; [10] has proposed $G$-convolution beyond the conventional translational equivariance of convolutional neural networks. [12] has generalized it further for Lie groups. $G$-orbit pooling, proposed by [23], can be applied to both arbitrary discrete groups and continuous groups by discretizing them. Some research has attempted to discover the underlying invariances within data [4, 46, 31, 35]. These approaches can be integrated as a major component of auto-encoders to represent the quotient symmetry of the data manifold, as we will discuss later in Section 3.3.

More relevant to our work, [37] has proposed using the consistency regularization for VAEs to encourage consistency in the latent variables of two identical images with different poses. It can be considered as the unsupervised learning version of the data augmentation and does not guarantee the explicit $G$-invariance of the regularized latent space. [5] has disentangled image content from pose variables, e.g., rotations or translations under the VAE framework. However, it has some limitations,

such as the requirement of using a specific decoder, the spatial transformer network [20]. It also has a problem that it cannot handle arbitrary groups beyond the rotation and translation. [11] has dealt with more complicated SO(3) group under the VAE framework. [29] has proposed the quotient auto-encoder (QAE) that can learn the quotient space of observations with respect to a given arbitrary group, extending early works in quotient images [34, 41]. However, it is a deterministic model and therefore cannot generate new samples easily. In addition, all the above-mentioned studies have not considered the Riemannian geometry of the learned latent representation.

## 2.2 Rimeannian Geometry of Generative Models

Recent studies have proposed to view the latent space of generative models as a Riemannian manifold [1, 9, 7]. In this perspective, the metric tensor of the latent space is given by a pullback metric induced from the observation space through the generator function. In line with this theoretical perspective, the flat manifold VAE (FMVAE) [8] and its variant [25, 15] have been proposed to regularize the generator function to be a Riemannian isometry by matching the pullback metric with the Euclidean one. The regularized latent space preserves geodesic distances of the observation space, making it a more geometrically meaningful representation that can be useful. However, the FMVAE and its variants do not take into account the underlying group symmetry, which can lead to inaccurate estimation of distances between data points. For example, they could estimate a non-zero distance between two identical images with different rotational angles, which is unsuitable for clustering tasks.

## 3 Method

Suppose $\mathbf{x} \in \mathcal{X} \subset \mathbb{R}^d$ is an observation where $\mathcal{X}$ is an extrinsic view of $\mathcal{M}$, i.e., the data manifold is realized via a mapping $\pi : \mathcal{M} \to \mathbb{R}^d$ such that $\mathcal{X} = \pi(\mathcal{M})$, and $\mathbf{z} \in \mathcal{Z} \subset \mathbb{R}^n$ is its latent representation where $n \ll d$. In addition, suppose $G$ to be a group (of symmetry transformations) that its group elements $g \in G$ naturally act on $\mathcal{X}$ by a left group action $\alpha : G \times \mathcal{X} \to \mathcal{X}$. We will denote $\alpha(g, \mathbf{x})$ as $g * \mathbf{x}$. Typical examples of $G$ include the special orthogonal group SO(2), a group of 2-dimensional rotations that can also naturally act on $d$-dimensional images by rotating them.

### 3.1 Auto-Encoders

The auto-encoder framework consists of two parameterized neural networks, an encoder $\mu_\theta : \mathbb{R}^d \to \mathbb{R}^n$ and a decoder $\mu_\phi : \mathbb{R}^n \to \mathbb{R}^d$. Vanilla auto-encoders find a low-dimensional compression of observations by minimizing the following reconstruction loss:

$$\mathbb{E}_{\mathbf{x} \sim p_{\mathcal{X}}(\mathbf{x})}\big[\mathcal{L}_{\mathrm{AE}}(\theta, \phi; \mathbf{x})\big] = \mathbb{E}_{\mathbf{x} \sim p_{\mathcal{X}}(\mathbf{x})}\big[\|\mathbf{x} - \mu_\phi \circ \mu_\theta(\mathbf{x})\|_2^2\big], \tag{1}$$

where $p_{\mathcal{X}}(\mathbf{x})$ is the data distribution. The expectation over $p_{\mathcal{X}}(\mathbf{x})$ can be computed via Monte Carlo (MC) estimation with finite samples $\mathrm{X} = \{\mathbf{x}_i\}_{i=1}^N$ as $\mathbb{E}_{p_{\mathcal{X}}(\mathbf{x})}\big[\mathcal{L}_{\mathrm{AE}}(\theta, \phi; \mathbf{x})\big] \approx (1/N)\sum_{i=1}^N \|\mathbf{x}_i - \mu_\phi \circ \mu_\theta(\mathbf{x}_i)\|_2^2$. Unless otherwise mentioned, we omit the expectation over $p_{\mathcal{X}}(\mathbf{x})$ for brevity.

Although sufficiently deep auto-encoders can provide effective latent compression $\mu_\theta(\mathrm{X}) = \mathrm{Z} \subset \mathbb{R}^n$ of high-dimensional data, they tend to overfit and the learned latent manifolds are often inaccurate [25], i.e., they cannot consider the underlying geometry of data. This can lead to problems such as incorrect latent interpolation or poor performance in downstream tasks, in addition to the absence of the ability to generate new samples.

### 3.2 Variational Auto-Encoders

The VAE [22] is a stochastic auto-encoding architecture belonging to the families of deep generative models. Contrary to vanilla auto-encoders, the VAE framework consists of two parameterized distributions $q_\theta(\mathbf{z}|\mathbf{x})$ and $p_\phi(\mathbf{x}|\mathbf{z})$ where the former and the latter are variational posterior and likelihood, respectively. VAEs try to maximize the marginal log-likelihood of observations by optimizing the following evidence lower bound:

$$\log p_\phi(\mathbf{x}) \geq \mathbb{E}_{\mathbf{z} \sim q_\theta(\mathbf{z}|\mathbf{x})}\big[\log p_\phi(\mathbf{x}|\mathbf{z})\big] - D_{\mathrm{KL}}\big(q_\theta(\mathbf{z}|\mathbf{x})\|p_{\mathcal{Z}}(\mathbf{z})\big) \triangleq -\mathcal{L}_{\mathrm{VAE}}(\theta, \phi; \mathbf{x}), \tag{2}$$

where $p_{\mathcal{Z}}(\mathbf{z})$ is the latent prior and $D_{\mathrm{KL}}(\cdot, \cdot)$ is the Kullback-Leibler (KL) divergence. $p_{\mathcal{Z}}(\mathbf{z})$ is often given by a normal distribution $\mathcal{N}(\mathbf{z}|\mathbf{0}, \mathbf{I}_n)$ where $\mathbf{0}$ is a zero vector and $\mathbf{I}_n$ is $n \times n$ identity matrix.

For vanilla VAEs, $q_\theta(\mathbf{z}|\mathbf{x})$ is chosen as a multivariate Gaussian distribution with a diagonal covariance matrix $\mathcal{N}(\mathbf{z}|\mu_\theta(\mathbf{x}), \text{diag}[\sigma_\theta^2(\mathbf{x})])$ and represented by a neural network encoder $(\mu_\theta, \sigma_\theta) : \mathbb{R}^d \to \mathbb{R}^n \times \mathbb{R}_+^n$. A latent variable is sampled by using the reparameterization trick $\mathbf{z} = \mu_\theta(\mathbf{x}) + \sigma_\theta(\mathbf{x}) \odot \epsilon$ where $\epsilon \sim \mathcal{N}(\epsilon|\mathbf{0}, \mathbf{I}_n)$. Although $p_\phi(\mathbf{x}|\mathbf{z})$ is chosen depending on the modeling of the data, it is often taken as a simple distribution such as a Gaussian with fixed variance, $\mathcal{N}(\mathbf{x}|\mu_\phi(\mathbf{z}), \beta\mathbf{I}_d)$, represented by a neural network decoder $\mu_\phi : \mathbb{R}^n \to \mathbb{R}^d$. In this case, the VAE objective (2) can be seen as a regularized version of (1) given by a sum of the stochastic auto-encoding reconstruction term $\mathcal{L}_{\text{AE}}(\theta, \phi; \mathbf{x}, \epsilon) = \|\mathbf{x} - \mu_\phi(\mu_\theta(\mathbf{x}) + \sigma_\theta(\mathbf{x}) \odot \epsilon)\|_2^2$ and $\beta$-weighted KL divergence $\mathcal{L}_{\text{KL}}(\theta; \mathbf{x}, \beta) = \beta D_{\text{KL}}(\mathcal{N}(\mathbf{z}|\mu_\theta(\mathbf{x}), \text{diag}[\sigma_\theta^2(\mathbf{x})])\|\mathcal{N}(\mathbf{z}|\mathbf{0}, \mathbf{I}_n))$ as follows:

$$\mathcal{L}_{\text{VAE}}(\theta, \phi; \mathbf{x}, \beta) = \mathbb{E}_{\epsilon \sim \mathcal{N}(\epsilon)}\big[\mathcal{L}_{\text{AE}}(\theta, \phi; \mathbf{x}, \epsilon)\big] + \mathcal{L}_{\text{KL}}(\theta; \mathbf{x}, \beta). \tag{3}$$

Practically, the expectation over $\mathcal{N}(\epsilon)$ term is estimated via a single-point MC when the mini-batch size is sufficiently large [22]. When $\beta \to 0$ and $\sigma_\theta^2 \to 0$, VAEs reduce to vanilla auto-encoders.

By minimizing (3), VAEs can learn the probabilistic process on the smooth latent space [13] that can easily generate plausible new samples by sampling a latent variable and decoding it, i.e., $\mu_\phi(\mathcal{Z}) \approx \mathcal{X}$. However, VAEs, like vanilla auto-encoders, do not ensure that the learned latent representation preserves the crucial information about the symmetry and geometry of the data manifold.

### 3.3 Quotient Auto-Encoders

QAE [29] is a modification of the deterministic auto-encoding framework that can find the quotient of the observation set $X/G$ by replacing (1) as the following quotient reconstruction loss:

$$\mathcal{L}_{\text{QAE}}(\theta, \phi; \mathbf{x}, G) = \inf_{g \in G} \|g * \mathbf{x} - \mu_\phi \circ \mu_\theta(\mathbf{x})\|_2^2. \tag{4}$$

As shown in (4), QAEs aim to minimize the set distance between the auto-encoded sample $\mu_\phi \circ \mu_\theta(\mathbf{x})$ and $G$-orbit $G * \mathbf{x} = \{g * \mathbf{x} | g \in G\}$ of the given $\mathbf{x}$ rather than $\mathbf{x}$ itself. It is noteworthy that $G$-orbit loses the information of $G$ for a given $\mathbf{x}$, which is the case of the natural quotient for $\mathbf{x}$ up to $G$. Thus, QAE can learn the quotient space of observations directly: for the ideal case of QAEs, i.e., when $\mathcal{L}_{\text{QAE}}(\theta, \phi; \mathbf{x}, G) \to 0$, the following property holds:

$$\mu_\phi \circ \mu_\theta(\mathbf{x}) = \mu_\phi \circ \mu_\theta(\mathbf{x} = g * \mathbf{x}) \triangleq \hat{\mathbf{x}}, \quad \forall g \in G,$$

which means that the image $\mu_\phi \circ \mu_\theta(X) \triangleq \hat{X}$ does not contain any information on $g$. In addition, QAEs typically adopt $G$-invariant architectures such as $G$-orbit pooling [23] (see Section B of Supplementary Material (SM) for details) for the encoder part to explicitly guarantee the following $G$-invariant latent representation:

$$\mu_\theta^G(\mathbf{x}) = \mu_\theta^G(g * \mathbf{x}) \triangleq \hat{\mathbf{z}}, \quad \forall g \in G,$$

where $\mu_\theta^G : \mathbb{R}^d \to \mathbb{R}^n$ is the $G$-invariant encoder. It is worth mentioning that these encoder architectures are unable to learn a meaningful representation of the data when using a vanilla auto-encoding framework, as the reconstruction loss (1) cannot be reduced enough; on the other hand, QAEs can use the explicitly $G$-invariant encoding architectures easily by taking advantage of the infimum in (4). This allows for both $\hat{X}$ and $\mu_\theta^G(X) \triangleq \hat{Z}$ to possess the invariant property up to $G$.

### 3.4 Isometric Quotient Variational Auto-Encoders

Although QAEs with $G$-invariant encoders can extract the underlying symmetry of observations, they still have the following drawbacks. First, QAEs are deterministic and may overfit to limited observations, similar to vanilla auto-encoders. Second, the Riemannian geometry of the extracted quotient manifold may not be preserved in the latent space. To address these issues, we present a stochastic version of QAEs called QVAEs, and further improve them with a novel framework for isometric learning called IQVAEs.

**QVAEs.** Inspired by the connection among (1), (3), and (4), we propose the stochastic version of QAEs (QVAEs) which minimizes the following stochastic quotient objective:

$$\mathcal{L}_{\text{QVAE}}(\theta, \phi; \mathbf{x}, \beta, G) \triangleq \mathbb{E}_{\epsilon \sim \mathcal{N}(\epsilon)}\big[\mathcal{L}_{\text{QAE}}(\theta, \phi; \mathbf{x}, \epsilon, G)\big] + \mathcal{L}_{\text{KL}}(\theta; \mathbf{x}, \beta, G), \tag{5}$$

where the first term is defined as $\mathcal{L}_{\mathrm{QAE}}(\theta, \phi; \mathbf{x}, \epsilon, G) \triangleq \inf_{g \in G} \| g * \mathbf{x} - \mu_\phi(\mu_\theta^G(\mathbf{x}) + \sigma_\theta^G(\mathbf{x}) \odot \epsilon) \|_2^2$ and the second term is $\mathcal{L}_{\mathrm{KL}}(\theta; \mathbf{x}, \beta, G) \triangleq \beta D_{\mathrm{KL}}\big(\mathcal{N}(\hat{\mathbf{z}}|\mu_\theta^G(\mathbf{x}), \mathrm{diag}[\sigma_\theta^G(\mathbf{x})]^2) \| p_{\mathcal{Z}}(\hat{\mathbf{z}})\big)$. We denote $(\mu_\theta^G, \sigma_\theta^G) : \mathbb{R}^d \to \mathbb{R}^n \times \mathbb{R}_+^n$ for the $G$-invariant multivariate Gaussian encoder.

QVAEs are able to generate new samples from the quotient space, unlike QAEs. It means that QVAEs can approximate the quotient space of the data manifold $\mathcal{X}/G \triangleq \hat{\mathcal{X}}$ as a form of the immersed manifold $\mu_\phi(\hat{\mathcal{Z}}) \approx \hat{\mathcal{X}}$ where $\hat{\mathcal{Z}}$ is a $G$-invariant latent representation. This leads us to introduce the concept of Riemannian isometry as a geometric regularization for QVAEs.

**Isometric regularziation to quotient spaces.** Before introducing the isometric regularization of the proposed QVAE, we review some basic concepts of the Riemannian geometry [2, 25].

**Definition 3.1. (Riemannian manifolds)** A $n$-dimensional Riemannian manifold $\mathcal{X}$ is a $n$-dimensional differentiable manifold that is equipped with a symmetric positive-definite metric tensor $\mathbf{H}_\mathcal{X}(\mathbf{x}) \in \mathbb{R}^{n \times n}$ ($n \times n$ matrix). It defines a local smooth inner product on the tangent space $\mathcal{T}_\mathbf{x}\mathcal{X}$ as $\langle \mathbf{u}_\mathbf{x}, \mathbf{v}_\mathbf{x} \rangle_{\mathbf{H}_\mathcal{X}} = \mathbf{u}_\mathbf{x}^\mathrm{T}\mathbf{H}_\mathcal{X}(\mathbf{x})\mathbf{v}_\mathbf{x}$ with tangent vectors $\mathbf{u}_\mathbf{x}, \mathbf{v}_\mathbf{x} \in \mathcal{T}_\mathbf{x}\mathcal{X}$ at each $\mathbf{x} \in \mathcal{X}$, allowing us to measure the distance and angles between tangent vectors at each point.

**Definition 3.2. (Riemannian distances)** Suppose $\mathcal{X}$ is a connected Riemannian manifold equipped with a Riemannian metric $\mathbf{H}_\mathcal{X}(\mathbf{x})$ at each point $\mathbf{x} \in \mathcal{X}$. A length of a curve on $\mathcal{X}$ between two points $\mathbf{x}_0$ and $\mathbf{x}_1$ is defined as:

$$L[\gamma_\mathcal{X}] = \int_0^1 dt \sqrt{\langle \dot{\gamma}_\mathcal{X}(t), \dot{\gamma}_\mathcal{X}(t) \rangle_{\mathbf{H}_\mathcal{X}}} = \int_0^1 dt \sqrt{\dot{\gamma}_\mathcal{X}^\mathrm{T}(t)\mathbf{H}_\mathcal{X}(\gamma_\mathcal{X}(t))\dot{\gamma}_\mathcal{X}(t)},$$

where $\gamma_\mathcal{X} : [0, 1] \to \mathcal{X}$ is a curve on $\mathcal{X}$ that travels from $\mathbf{x}_0 = \gamma_\mathcal{X}(0)$ to $\mathbf{x}_1 = \gamma_\mathcal{X}(1)$ and $\dot{\gamma}_\mathcal{X}(t) = \partial_t \gamma_\mathcal{X}(t) \in \mathcal{T}_{\gamma_\mathcal{X}(t)}\mathcal{X}$ is the velocity of the curve. The curve minimizing $L$ is called a geodesic. The Riemannian distance is given by the length of the geodesic, i.e., $d_\mathcal{X}(\mathbf{x}_0, \mathbf{x}_1) = \inf_{\gamma_\mathcal{X}} L[\gamma_\mathcal{X}]$.

**Definition 3.3. (Immersions)** Let $\mathcal{Z}$ be an open set in $\mathbb{R}^n$. A smooth map $\mu : \mathcal{Z} \to \mathbb{R}^d$ ($n < d$) is called an immersion if its pushforward $d\mu_\mathbf{z} : \mathcal{T}_\mathbf{z}\mathcal{Z} \to \mathcal{T}_{\mu(\mathbf{z})}\mu(\mathcal{Z})$ is injective for all $\mathbf{z} \in \mathcal{Z}$. Then, $\mu(\mathcal{Z})$ is called an immersed submanifold.

**Definition 3.4. (Geometry of immersed submanifolds)** Suppose $\mu : \mathcal{Z} \to \mathbb{R}^d$ is an immersion onto a submanifold $\mathcal{X} \subset \mathbb{R}^d$. The Euclidean inner product in $\mathbb{R}^d$ is a proper inner product of tangent vectors of $\mathcal{X}$. Thus, $\mathcal{X}$ is a Riemannian manifold equipped with a Riemannian metric $\mathbf{H}_\mathcal{X}(\mathbf{x}) = \mathbf{I}_d$, i.e., $\langle \mathbf{u}_\mathbf{x}, \mathbf{v}_\mathbf{x} \rangle_{\mathbf{H}_\mathcal{X}} = \mathbf{u}_\mathbf{x}^\mathrm{T}\mathbf{I}_d\mathbf{v}_\mathbf{x} = \mathbf{u}_\mathbf{x}^\mathrm{T}\mathbf{v}_\mathbf{x}$ for tangent vectors $\mathbf{u}_\mathbf{x}, \mathbf{v}_\mathbf{x} \in \mathcal{T}_\mathbf{x}\mathcal{X}$ at each $\mathbf{x} \in \mathcal{X}$.

**Definition 3.5. (Pullback metric)** Suppose $\mu : \mathcal{Z} \to \mathbb{R}^d$ is an immersion onto a Riemannian submanifold $\mathcal{X} \subset \mathbb{R}^d$ equipped with a Riemannian metric $\mathbf{I}_d$. Then, for tangent vectors $\mathbf{u}_\mathbf{x}, \mathbf{v}_\mathbf{x} \in \mathcal{T}_\mathbf{x}\mathcal{X}$ at each $\mathbf{x} = \mu(\mathbf{z}) \in \mathcal{X} = \mu(\mathcal{Z})$, the following holds:

$$\langle \mathbf{u}_\mathbf{x}, \mathbf{v}_\mathbf{x} \rangle_{\mathbf{H}_\mathcal{X} = \mathbf{I}_d} = \mathbf{u}_\mathbf{x}^\mathrm{T}\mathbf{v}_\mathbf{x} = (d\mu_\mathbf{z}\mathbf{u}_\mathbf{z})^\mathrm{T}d\mu_\mathbf{z}\mathbf{v}_\mathbf{z} = \mathbf{u}_\mathbf{z}^\mathrm{T}[(d\mu_\mathbf{z})^\mathrm{T}d\mu_\mathbf{z}]\mathbf{v}_\mathbf{z} \triangleq \mathbf{u}_\mathbf{z}^\mathrm{T}\mathbf{H}_\mu(\mathbf{z})\mathbf{v}_\mathbf{z} = \langle \mathbf{u}_\mathbf{z}, \mathbf{v}_\mathbf{z} \rangle_{\mathbf{H}_\mu}.$$

Therefore, $\mathcal{Z}$ can be viewed as a Riemannian manifold equipped with $\mathbf{H}_\mu(\mathbf{z})$ called a pullback metric. In addition, because $\mathcal{Z} \subset \mathbb{R}^n$ and $\mathcal{X} \subset \mathbb{R}^d$, the pushforward $d\mu_\mathbf{z}$ is equal to the standard Jacobian $\mathbf{J}_\mu(\mathbf{z})$ of $\mu$, i.e., $\mathbf{H}_\mu(\mathbf{z}) = \mathbf{J}_\mu^\mathrm{T}(\mathbf{z})\mathbf{J}_\mu(\mathbf{z})$.

**Definition 3.6. (Isometric immersion)** Suppose $\mathcal{Z}$ is a Riemannian manifold equipped with a Riemannian metric $\mathbf{H}_\mathcal{Z}(\mathbf{z})$ at each $\mathbf{z} \in \mathcal{Z}$. A smooth immersion $\mu : \mathcal{Z} \to \mathbb{R}^d$ onto $\mathcal{X} \subset \mathbb{R}^d$ is a Riemannian isometry if the metric $\mathbf{H}_\mathcal{Z}(\mathbf{z})$ is equal to the pullback metric $\mathbf{H}_\mu(\mathbf{z})$, i.e., $\mathbf{H}_\mathcal{Z}(\mathbf{z}) = \mathbf{J}_\mu^\mathrm{T}(\mathbf{z})\mathbf{J}_\mu(\mathbf{z})$ for every $\mathbf{z} \in \mathcal{Z}$. In this case, the length of curves on $\mathcal{X}$ is equivalent with that on $\mathcal{Z}$:

$$L[\gamma_\mathcal{X}] = \int_0^1 dt \sqrt{\langle \dot{\gamma}_\mathcal{X}(t), \dot{\gamma}_\mathcal{X}(t) \rangle} = \int_0^1 dt \sqrt{\dot{\gamma}_\mathcal{X}^\mathrm{T}(t)\dot{\gamma}_\mathcal{X}(t)} = \int_0^1 dt \sqrt{\dot{\gamma}_\mathcal{Z}^\mathrm{T}(t)\mathbf{J}_\mu^\mathrm{T}(\gamma_\mathcal{Z}(t))\mathbf{J}_\mu(\gamma_\mathcal{Z}(t))\dot{\gamma}_\mathcal{Z}(t)}$$

$$= \int_0^1 dt \sqrt{\dot{\gamma}_\mathcal{Z}^\mathrm{T}(t)\mathbf{H}_\mathcal{Z}(\gamma_\mathcal{Z}(t))\dot{\gamma}_\mathcal{Z}(t)} = \int_0^1 dt \sqrt{\langle \dot{\gamma}_\mathcal{Z}(t), \dot{\gamma}_\mathcal{Z}(t) \rangle_{\mathbf{H}_\mathcal{Z}}} = L[\gamma_\mathcal{Z}],$$

where $\gamma_\mathcal{Z} : [0, 1] \to \mathcal{Z}$ and $\gamma_\mathcal{X} : [0, 1] \to \mathcal{X}$ are respectively curves from $\mathbf{z}_0 = \gamma_\mathcal{Z}(0)$ to $\mathbf{z}_1 = \gamma_\mathcal{Z}(1)$ and from $\mathbf{x}_0 = \gamma_\mathcal{X}(0)$ to $\mathbf{x}_1 = \gamma_\mathcal{X}(1)$, when $\mathbf{x}_0 = \mu(\mathbf{z}_0)$, $\mathbf{x}_1 = \mu(\mathbf{z}_1)$, $\dot{\gamma}_\mathcal{Z}(t) \in \mathcal{T}_{\gamma_\mathcal{Z}(t)}\mathcal{Z}$, and accordingly $\dot{\gamma}_\mathcal{X}(t) \in \mathcal{T}_{\mu(\gamma_\mathcal{Z}(t))}\mu(\mathcal{Z})$.

With some mild assumptions [36], a sufficiently well-trained decoder $\mu_\phi : \hat{\mathcal{Z}} \to \mathbb{R}^d$ of the QVAE can be viewed as a smooth immersion onto the quotient of $\hat{\mathcal{X}} = \mathcal{X}/G$. Then, from Definition 3.6,

the following regularization should be added to (5) to make the decoder be a Riemannian isometry between the latent representation $\hat{\mathcal{Z}}$ and quotient $\hat{\mathcal{X}}$:

$$\mathcal{L}_{\text{ISO}}(\theta, \phi; \mathbf{x}, \mathbf{H}_{\hat{\mathcal{Z}}}, \lambda) = \lambda \mathbb{E}_{\hat{\mathbf{z}} \sim q_\theta(\hat{\mathbf{z}}|\mathbf{x})} \|\mathbf{J}_{\mu_\phi}^{\text{T}}(\hat{\mathbf{z}})\mathbf{J}_{\mu_\phi}(\hat{\mathbf{z}}) - \mathbf{H}_{\hat{\mathcal{Z}}}(\hat{\mathbf{z}})\|_F, \tag{6}$$

where $\mathbf{J}_{\mu_\phi}(\hat{\mathbf{z}})$ is Jacobian of $\mu_\phi$ at $\hat{\mathbf{z}}$, $\lambda$ is a hyper-parameter, and $\mathbf{H}_{\hat{\mathcal{Z}}}(\hat{\mathbf{z}})$ is a metric tensor of $\hat{\mathcal{Z}}$.

There are two things that should be clarified for practical computation of (6). The first aspect to consider is the selection of the Riemannian metric $\mathbf{H}_{\hat{\mathcal{Z}}}(\hat{\mathbf{z}})$. A commonly used and particularly useful choice of $\mathbf{H}_{\hat{\mathcal{Z}}}(\hat{\mathbf{z}})$ is the scaled Euclidean metric in $\mathbb{R}^n$, i.e., $\mathbf{H}_{\hat{\mathcal{Z}}}(\hat{\mathbf{z}}) = \mathbf{C}_n = c^2\mathbf{I}_n$ where $c$ is a constant. This is because it is the case that using the Euclidean distance[1] on the latent representation $d_{\hat{\mathcal{Z}}}(\hat{\mathbf{z}}_0, \hat{\mathbf{z}}_1) = \sqrt{(\hat{\mathbf{z}}_1 - \hat{\mathbf{z}}_0)^{\text{T}}\mathbf{I}_n(\hat{\mathbf{z}}_1 - \hat{\mathbf{z}}_0)}$ can preserve the geodesic distance on the quotient manifold $d_{\hat{\mathcal{X}}}(\hat{\mathbf{x}}_0, \hat{\mathbf{x}}_1)$ up to $c$. The constant $c$ can be regarded as either a pre-defined hyper-parameter or a learnable one. More generally, one can use any parameterized symmetric positive-definite matrix $\mathbf{C}_n$ for $\mathbf{H}_{\hat{\mathcal{Z}}}(\hat{\mathbf{z}})$. Note that such matrices can be achieved from an arbitrary learnable parameterized $n \times n$ matrix $\mathbf{M}_n$ by using the Cholesky decomposition as follows:

---

**Algorithm 1** IQVAEs

**Input:** data $\{\mathbf{x}_i\}_{i=1}^N$, hyper-parameters $(\beta, \lambda)$, group $G$, $G$-invariant encoders $(\mu_\theta^G, \sigma_\theta^G)$, decoder $\mu_\phi$
Initialize $\theta, \phi, \mathbf{C}_n$
**while** training **do**
    Sample $\{\alpha_i \sim [0,1]\}_i^N$, $\{\epsilon_i \sim \mathcal{N}(\mathbf{0},\mathbf{I})\}_i^N$
    Compute $\{\mu_\theta^i, \sigma_\theta^i\}_i^N = \{\mu_\theta^G(\mathbf{x}_i), \sigma_\theta^G(\mathbf{x}_i)\}_i^N$
    Sample $\{\mathbf{z}_i\}_{i=1}^N = \{\mu_\theta^i + \sigma_\theta^i \odot \epsilon_i\}_{i=1}^N$
    Shuffle $\{\mathbf{z}_j\}_{j=1}^N = \texttt{shuffle}(\{\mathbf{z}_i\}_{i=1}^N)$
    Augment $\{\tilde{\mathbf{z}}_i\}_{i=1}^N = \{(1-\alpha_i)\mathbf{z}_i + \alpha_i\mathbf{z}_j\}_{i=j=0}^{i=j=N}$
    Compute $\mathcal{L}_{\text{QAE}} = \sum_{i=1}^N \min_{g \in G} \|g * \mathbf{x}_i - \mu_\phi(\mathbf{z}_i)\|_2^2$
    Compute $\mathcal{L}_{\text{KL}} = \sum_{i=1}^N D_{\text{KL}}(\mathcal{N}(\mu_\theta^i, \text{diag}[\sigma_\theta^i]^2)\|\mathcal{N}(\mathbf{0},\mathbf{I}_n))$
    Compute $\{\mathbf{J}_{\mu_\phi}^i\}_{i=1}^N = \{\mathbf{J}_{\mu_\phi}(\tilde{\mathbf{z}}_i)\}_i^N$
    Compute $\mathcal{L}_{\text{ISO}} = \sum_{i=1}^N \|(\mathbf{J}_{\mu_\phi}^i)^{\text{T}}\mathbf{J}_{\mu_\phi}^i - \mathbf{C}_n\|_F$
    Optimize $(\mathcal{L}_{\text{QAE}} + \beta\mathcal{L}_{\text{KL}} + \lambda\mathcal{L}_{\text{ISO}})/N$ w.r.t $\theta, \phi$
**end while**

---

$$\mathbf{C}_n = \mathbf{L}_n(\mathbf{L}_n)^{\text{T}}, \quad \mathbf{L}_n = \text{lower}[\mathbf{M}_n] - \text{diag}[\mathbf{M}_n] + \text{diag}[\mathbf{M}_n]^2 + \varepsilon^2\mathbf{I}_n, \tag{7}$$

where $\mathbf{L}_n$ is a real lower triangular matrix with all positive diagonal entries and $\varepsilon$ is a small non-zero constant. In this case, using the Mahalanobis distance on the latent space $d_{\hat{\mathcal{Z}}}(\hat{\mathbf{z}}_0, \hat{\mathbf{z}}_1) = \sqrt{(\hat{\mathbf{z}}_1 - \hat{\mathbf{z}}_0)^{\text{T}}\mathbf{C}_n(\hat{\mathbf{z}}_1 - \hat{\mathbf{z}}_0)}$ preserves the geodesic distance of the quotient space of the observation manifold $d_{\hat{\mathcal{X}}}(\hat{\mathbf{x}}_0, \hat{\mathbf{x}}_1)$.

The second aspect to consider is the sampling from $q_\theta(\hat{\mathbf{z}}|\mathbf{x})$. It can be considered as Gaussian vicinal distribution for finite observation samples that smoothly fills the latent space where data is missing. In addition to that, following [8], we use the mix-up vicinal distribution [39] to effectively regularize the entire space of interest to be isometric. As a result, the tractable form of (6) is equal to[2]:

$$\lambda \mathbb{E}_{\mathbf{x}_{i,j} \sim p_{\mathcal{X}}, \epsilon_{i,j} \sim \mathcal{N}, \alpha \sim [0,1]} \|\mathbf{J}_{\mu_\phi}^{\text{T}}(f_\theta^\alpha(\mathbf{x}_{i,j}, \epsilon_{i,j}))\mathbf{J}_{\mu_\phi}(f_\theta^\alpha(\mathbf{x}_{i,j}, \epsilon_{i,j})) - \mathbf{C}_n\|_F, \tag{8}$$

where $f_\theta^\alpha(\mathbf{x}_{i,j}, \epsilon_{i,j}) = (1-\alpha)\hat{\mathbf{z}}_i + \alpha\hat{\mathbf{z}}_j$ is the latent mix-up for $\hat{\mathbf{z}}_{i,j} = \mu_\theta^G(\mathbf{x}_{i,j}) + \sigma_\theta^G(\mathbf{x}_{i,j}) \odot \epsilon_{i,j}$. Practically, it is computed by sampling a mini-batch of latent variables, i.e., $\{\hat{\mathbf{z}}_i\}_{i=1}^N = \{\mu_\theta^G(\mathbf{x}_i) + \sigma_\theta^G(\mathbf{x}_i) \odot \epsilon_i\}_{i=1}^N$, shuffling it for $\{\hat{\mathbf{z}}_j\}_{j=1}^N = \texttt{shuffle}(\{\hat{\mathbf{z}}_i\}_{i=1}^N)$, and mixing-up them.

**IQVAE.** We define the IQVAE as a class of QVAEs whose objective function is given by the sum of the variational quotient objective $\mathcal{L}_{\text{QVAE}}$ (5) and Riemannian isometry $\mathcal{L}_{\text{ISO}}$ (8) as follows:

$$\mathcal{L}_{\text{IQVAE}}(\theta, \phi; \mathbf{x}, \beta, \lambda, G) \triangleq \mathcal{L}_{\text{QVAE}}(\theta, \phi; \mathbf{x}, \beta, G) + \mathcal{L}_{\text{ISO}}(\theta, \phi, \mathbf{C}_n; \mathbf{x}, \lambda). \tag{9}$$

The proposed IQVAE optimization procedure is summarized in Algorithm 1 (see Section A of SM for additional tips). In the IQVAE, we typically set $\mathbf{C}_n = c^2\mathbf{I}_n$. For clarity, we will refer to the IQVAE with (7) as IQVAE-M, to distinguish it from the basic version.

## 4 Experiments

We compared our proposed QVAE and IQVAE with six different competitors: the auto-encoder (AE), VAE, $\beta$-VAE [19], consistency regularized VAE (CRVAE) [37], QAE [29], and FMVAE [8]. We used the same convolutional architecture[3], hyper-parameters, optimizer [21], and training scheme for

---

[1]We assume the straight path between $\hat{\mathbf{z}}_0$ and $\hat{\mathbf{z}}_1$ is an on-manifold path for every pair $\hat{\mathbf{z}}_0, \hat{\mathbf{z}}_1 \in \hat{\mathcal{Z}}$.

[2]In (8), we temporarily retrieve the expectation on observations to make things more explicit.

[3]We used $G$-orbit pooling (see Section B of SM for details), which can achieve $G$-invariance in combination with arbitrary convolutional models, for encoders of QAE, QVAE, and IQVAE.

Table 1: The averaged clustering ARIs and classification test accuracies for $(2\pi/3)$-rotated (left) and $2\pi$-rotated (right) MNIST across various models, each repeated five times.

| METHOD | $k$-MEANS | GMM | SVM | RF | METHOD | $k$-MEANS | GMM | SVM | RF |
|---|---|---|---|---|---|---|---|---|---|
| AE | $21.8_{\pm2.3}$ | $29.5_{\pm2.2}$ | $86.4_{\pm0.4}$ | $86.2_{\pm0.5}$ | AE | $10.0_{\pm1.0}$ | $12.5_{\pm2.0}$ | $68.3_{\pm2.6}$ | $71.2_{\pm2.1}$ |
| VAE | $23.1_{\pm0.5}$ | $28.4_{\pm0.7}$ | $87.4_{\pm0.4}$ | $87.4_{\pm0.4}$ | VAE | $11.4_{\pm1.2}$ | $14.7_{\pm1.6}$ | $72.8_{\pm1.3}$ | $73.3_{\pm0.9}$ |
| $\beta$-VAE | $21.5_{\pm1.0}$ | $26.5_{\pm0.8}$ | $88.5_{\pm0.5}$ | $88.5_{\pm0.5}$ | $\beta$-VAE | $10.1_{\pm0.8}$ | $11.9_{\pm1.1}$ | $72.5_{\pm0.5}$ | $73.5_{\pm0.5}$ |
| CRVAE | $34.9_{\pm1.3}$ | $37.4_{\pm2.4}$ | $93.0_{\pm0.2}$ | $\mathbf{93.0_{\pm0.3}}$ | CRVAE | $26.1_{\pm2.7}$ | $33.4_{\pm3.6}$ | $78.3_{\pm2.4}$ | $80.3_{\pm2.0}$ |
| QAE | $45.2_{\pm4.2}$ | $65.5_{\pm3.6}$ | $\mathbf{93.1_{\pm0.3}}$ | $92.7_{\pm0.3}$ | QAE | $36.9_{\pm5.2}$ | $52.9_{\pm3.1}$ | $91.8_{\pm0.6}$ | $91.7_{\pm0.3}$ |
| FMVAE | $19.0_{\pm1.2}$ | $25.7_{\pm2.0}$ | $85.6_{\pm0.5}$ | $85.1_{\pm0.6}$ | FMVAE | $10.1_{\pm0.9}$ | $11.9_{\pm1.0}$ | $67.6_{\pm0.7}$ | $69.2_{\pm0.4}$ |
| QVAE | $59.3_{\pm6.5}$ | $72.6_{\pm5.2}$ | $93.0_{\pm0.4}$ | $92.5_{\pm0.5}$ | QVAE | $53.9_{\pm7.2}$ | $66.3_{\pm6.5}$ | $\mathbf{93.0_{\pm0.1}}$ | $92.5_{\pm0.1}$ |
| IQVAE | $\mathbf{72.8_{\pm2.9}}$ | $\mathbf{74.4_{\pm1.6}}$ | $\mathbf{93.2_{\pm0.2}}$ | $92.9_{\pm0.1}$ | IQVAE | $\mathbf{70.9_{\pm2.9}}$ | $\mathbf{72.9_{\pm1.5}}$ | $92.9_{\pm0.3}$ | $\mathbf{92.7_{\pm0.4}}$ |

all models. Implementation details can be found in Section D of SM. After training, we encoded all samples and extracted their latent representations across models. Then, we conducted latent-based downstream clustering tasks by using the simple $k$-means and Gaussian mixture model (GMM) [33]. Similarly, we conducted downstream classification tasks by using the support vector machine (SVM) [17] and random forest (RF) classifiers. We used the same hyper-parameters for downstream models regardless of the benchmarked latent representations. Then, we computed the adjusted Rand index (ARI) [38] and test classification accuracy for clustering and classification models, respectively.

## 4.1 Rotated MNIST

Rotated MNIST consists of randomly rotated images of handwritten digits. It has the rotational group symmetry of $G = SO(2)$. We padded the original MNIST images to size $32 \times 32$ for improved usability. Then, we generated $\varphi$-rotated MNIST by randomly rotating each sample in the MNIST with a uniform distribution of $[-\varphi/2, \varphi/2]$. We considered both $\varphi = 2\pi/3$ and $\varphi = 2\pi$ cases (see Section C of SM for examples). We used 60,000 samples for training and 10,000 samples for testing.

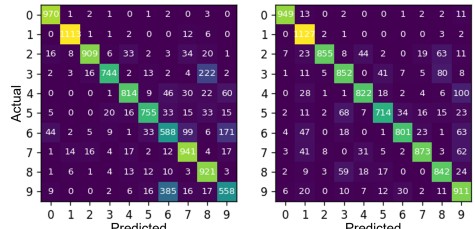

Figure 2: Clustering confusion matrices of (left) QVAE and (right) IQVAE for $\varphi = 2\pi$.

**Basic evaluation.** For quantitative analysis, we summarize downstream task performances across the models in Table 1. It clearly shows the proposed QVAE and IQVAE outperform other competitors by a large margin because they consider the group symmetry explicitly. Moreover, the IQVAE shows greatly improved clustering performances even over the QVAE when $\varphi = 2\pi$. We deduce that the Riemannian isometry helps the IQVAE recognize a meaning-

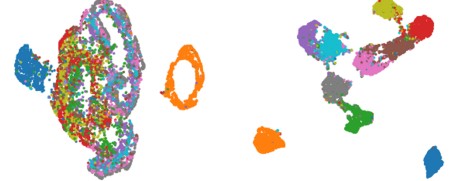

Figure 3: UMAP visualizations of (left) VAE and (right) IQVAE for $\varphi = 2\pi$.

ful distance relationship between controversial samples, e.g., digits 6 and 9, as depicted in Figure 2. For qualitative analysis, we also visualize the learned latent representations for the test dataset of the rotated MNIST by using the uniform manifold approximation and projection (UMAP) [28] (see Figure 3, and Section E of SM for all models). It further supports that the IQVAE succeeds in finding meaningful representations of the dataset.

**Aligned reconstructions.** Figure 4 shows six images of digit 7 and their reconstructions based on the proposed IQVAE demonstrating that the IQVAE can reconstruct rotated digits in the aligned frame thanks to learning the quotient up to rotations.

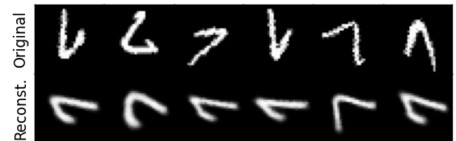

Figure 4: Six images of digit 7 and their reconstructions based on the proposed IQVAE.

**Better interpolation.** Figure 5 compares linear interpolations between digits 1 and 7 for the VAE, QVAE, and IQVAE. It is conducted by parameterizing a linear path as $\mathbf{z}(t) = \mathbf{z}(0) + (\mathbf{z}(10) - \mathbf{z}(0)) \cdot (t/10)$ and then reconstructing $\mathbf{x}(t) = \mu_\phi(\mathbf{z}(t))$ for $t \in \{0, 1, ..., 10\}$. As shown in Figure 5 (a), the VAE shows a meaningless interpolation from $t = 5$ to $t = 9$ (see Section F of SM for other examples).

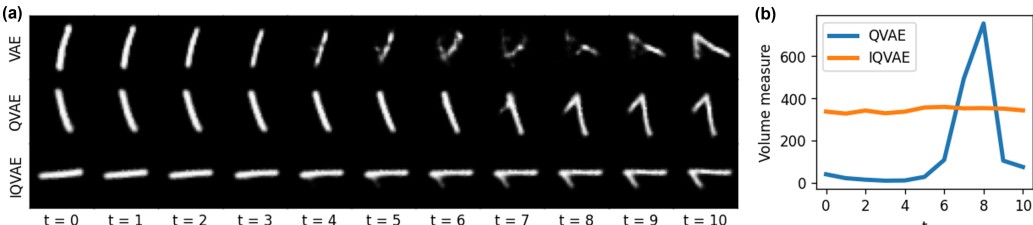

Figure 5: (a) Linear interpolations between digits 1 and 7 for the VAE, QVAE, and IQVAE. (b) The geometric volume measures along each linear path of the QVAE and IQVAE.

While the QVAE shows a more convincing interpolation compared to the VAE, it also shows abrupt changes at $t = 7$ and $t = 8$. The IQVAE shows the smoothest interpolation between digits 1 and 7 compared to the other models owing to the isometry regularization. The smooth interpolation of the IQVAE is also quantitatively confirmed in Figure 5 (b) which compares the geometric volume measures along each linear path of the QVAE and IQVAE. The volume measure is computed via $\sqrt{\det \mathbf{J}_{\mu_\phi}^{\mathrm{T}}(\mathbf{z}(t))\mathbf{J}_{\mu_\phi}(\mathbf{z}(t))}$ and can be viewed as a local infinitesimal volume ratio between the latent and observation spaces. The volume measure of the QVAE shows a clear peak near $t = 8$, while that of the IQVAE is nearly constant.

**Evidence for isometry.** To demonstrate that the learned decoding function of IQVAEs upholds the desired Riemannian isometry, we evaluated the condition numbers of the pull-back metric tensors and the Riemannian volume measures for all test samples across models. The condition number is defined as the ratio $\lambda_{\max}/\lambda_{\min}$, where $\lambda_{\max}$ and $\lambda_{\min}$ respectively represent the maximum and minimum eigenvalues of the pull-back metric, denoted as $\mathbf{J}_{\mu_\phi}^{\mathrm{T}}(\mathbf{z})\mathbf{J}_{\mu_\phi}(\mathbf{z})$. All computed volume measures

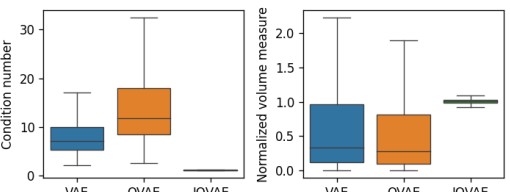

Figure 6: (left) Condition numbers and (right) volume measures for the $2\pi$-rotated MNIST for the VAE, QVAE, and IQVAE.

are normalized by their average values. Therefore, condition numbers and normalized volume measures approaching 1.0, accompanied by minimal variances, indicate that the learned latent representation is isometrically isomorphic to Euclidean space. In other words, the learned pull-back metric aligns with the Euclidean metric. As shown in Figure 6, the IQVAE presents condition numbers and volume measures close to 1.0, exhibiting trivial variances compared to other models.

**Sample efficiency comparison.** Table 2 compares the sample efficiency of IQVAE and the most comparable baseline, QAE, in terms of classification accuracy. The results demonstrate the effectiveness of the proposed method as IQVAE shows more robust performance with respect to variations in the training sample size.

Table 2: Classification test accuracies for $2\pi$-rotated MNIST with varying training sample sizes.

| SAMPLE SIZE | QAE + SVM | IQVAE + SVM |
|---|---|---|
| 5,000 | $83.9_{\pm 0.7}$ | $87.6_{\pm 0.6}$ |
| 10,000 | $85.5_{\pm 0.8}$ | $89.8_{\pm 1.0}$ |
| 60,000 | $91.8_{\pm 0.6}$ | $92.9_{\pm 0.3}$ |

**Test-time orbit augmentation.** The current QVAE and IQVAE require more computational cost than vanilla VAE due to the $G$-orbit pooling encoders, which augment the number of observations $|G|$ times larger by expanding a data point $\mathbf{x}$ as a $G$-orbit $G * \mathbf{x}$. It might be a bottleneck, especially when training the proposed model with larger-scale datasets. To resolve this issue, we suggest the test-time orbit augmentation strategy, which involves using a coarse step size when discretizing the group parameters (e.g., rotation angle) during training, and then switching to a finer step size during the inference phase. Table 3 compares wall-clock training

Table 3: Training wall-clock time per epoch and clustering ARIs of VAEs, QVAEs, and IQVAEs. We used a single V100 32GB GPU.

| METHOD | TIME [S] | $k$-MEANS | GMM |
|---|---|---|---|
| VAE | 1.12 | $11.4_{\pm 1.2}$ | $14.7_{\pm 1.6}$ |
| QVAE (36-36) | 10.17 | $53.9_{\pm 7.2}$ | $66.3_{\pm 6.5}$ |
| QVAE (12-12) | 4.06 | $50.8_{\pm 2.6}$ | $59.2_{\pm 1.5}$ |
| QVAE (12-36) | 4.06 | $54.6_{\pm 2.8}$ | $63.8_{\pm 3.9}$ |
| IQVAE (36-36) | 13.22 | $70.9_{\pm 2.9}$ | $72.9_{\pm 1.5}$ |
| IQVAE (12-12) | 6.11 | $59.6_{\pm 2.7}$ | $60.9_{\pm 1.8}$ |
| IQVAE (12-36) | 6.11 | $64.3_{\pm 2.1}$ | $65.0_{\pm 2.1}$ |

times and downstream task performances for $2\pi$-rotated MNIST datasets for the VAE, QVAE, and IQVAE. The former and latter numbers in parentheses for QVAEs and IQVAEs in the table indicate discretized group sizes used in the training and inference phases respectively. The results show that the proposed technique can scale the training time without severe performance degradation.

**Effect of mix-up.** We employed the mix-up technique (8) to regularize the entire latent space. By augmenting the latent space with the mix-up, it is possible to smoothly fill gaps in the latent space regions where data might be missing. This becomes particularly crucial when the number of available training samples is limited. To validate this, we trained IQVAEs both with and without the mix-up technique in two scenarios: one with a training sample size of 60,000 and the other with 10,000. Subsequently, we computed the condition numbers of the pull-back metric tensors and mean-normalized volume measures for test samples. Figure 7 shows that using the mix-up reduces the variances of both condition numbers and volume measures for pull-back metric tensors, demonstrating its efficacy in regularizing the entire latent space. Furthermore, the impact of the mix-up approach becomes more noticeable when the training sample size is smaller.

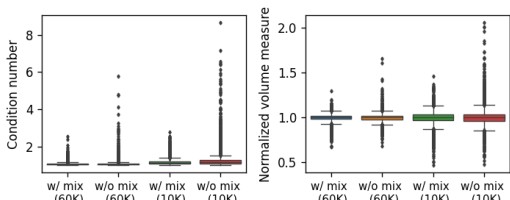

Figure 7: (left) Condition numbers of the pullback metrics and (right) volume measures of $2\pi$-rotated MNIST test samples, with and without the mix-up for IQVAEs. The numbers in parentheses indicate the training sample size.

## 4.2 MixedWM38

MixedWM38 is a dataset containing 38,015 $52 \times 52$ semiconductor wafer maps, separated into 38 defect categories including 1 normal type, 8 single-defect types, and 29 mixed-defect types [42] (see Section C of SM for examples). We assume rotation and reflection (i.e., $G = \mathrm{O}(2)$) symmetries are present in the MixedWM38. We used 30,720 samples for training and 7,295 samples for testing.

Table 4: Clustering ARIs and classification test F-scores for MixedWM38 (repeated five times).

| METHOD | $k$-MEANS | GMM | SVM | RF |
|---|---|---|---|---|
| AE | $18.7_{\pm0.8}$ | $22.0_{\pm0.6}$ | $81.7_{\pm0.6}$ | $83.8_{\pm0.2}$ |
| VAE | $21.3_{\pm0.7}$ | $24.9_{\pm0.6}$ | $82.5_{\pm0.3}$ | $84.1_{\pm0.5}$ |
| $\beta$-VAE | $21.3_{\pm0.4}$ | $27.1_{\pm1.1}$ | $82.5_{\pm0.6}$ | $84.5_{\pm0.5}$ |
| CRVAE | $21.4_{\pm0.6}$ | $24.6_{\pm0.6}$ | $82.3_{\pm0.9}$ | $84.9_{\pm1.0}$ |
| QAE | $22.7_{\pm0.9}$ | $30.7_{\pm1.5}$ | $84.4_{\pm0.3}$ | $85.1_{\pm0.2}$ |
| FMVAE | $17.6_{\pm0.4}$ | $25.8_{\pm0.7}$ | $81.4_{\pm0.2}$ | $83.1_{\pm0.4}$ |
| QVAE | $23.3_{\pm0.5}$ | $31.1_{\pm0.5}$ | $\mathbf{85.0_{\pm0.4}}$ | $85.1_{\pm0.1}$ |
| IQVAE | $\mathbf{27.5_{\pm0.2}}$ | $\mathbf{35.3_{\pm1.4}}$ | $84.4_{\pm0.2}$ | $\mathbf{85.6_{\pm0.3}}$ |

**Basic evaluation.** We performed an experiment on the MixedWM38 dataset, similar to the one previously conducted on rotated MNIST. Unlike the rotated MNIST, the MixedWM38 dataset requires a multi-label classification task, which is a key difference between the two experiments. Thus, we use the F-score as a classification accuracy for this problem. Table 4 summarizes the downstream task performances of eight models on the MixedWM38 dataset (see Section E of SM for UMAP visualizations). As shown in the table, the proposed methods have a higher performance on the downstream task than the other models.

**OoD evaluation.** We further demonstrate the effectiveness of the proposed IQVAE in detecting OoD samples, an important task in the semiconductor engineering field. In order to mimic a real-world setting, we divided the 7,015 single-defect type data (8 classes: C, D, EL, ER, L, NF, S, R; see Section C of SM) into two sets: C-EL-L-S as the in-distribution set, and D-ER-NF-R as the OoD set. Then, we randomly rotated all samples in the C-EL-L-S subset, and considered the rotated samples an in-distribution set, aligning with the nature of the semiconductor manufacturing process [43]. Using auto-encoding models trained on the original C-EL-L-S data, we formulated an OoD detection task, the goal of which is to differentiate between samples from the rotated C-EL-L-S set (in-distribution) and

Table 5: AUCROCs and AUPRCs for MixedWM38 across the models (repeated five times).

| METHOD | AUCROC | AUPRC |
|---|---|---|
| AE | $47.3_{\pm3.6}$ | $27.2_{\pm3.3}$ |
| VAE | $62.4_{\pm3.1}$ | $34.3_{\pm2.4}$ |
| $\beta$-VAE | $66.1_{\pm4.1}$ | $41.4_{\pm6.8}$ |
| CRVAE | $80.5_{\pm6.1}$ | $60.4_{\pm10.2}$ |
| QAE | $90.2_{\pm3.7}$ | $81.8_{\pm3.4}$ |
| FMVAE | $69.8_{\pm5.6}$ | $42.5_{\pm7.2}$ |
| QVAE | $90.3_{\pm1.1}$ | $79.5_{\pm3.9}$ |
| IQVAE | $\mathbf{93.8_{\pm0.4}}$ | $\mathbf{88.3_{\pm1.7}}$ |

samples from the D-ER-NF-R set (OoD). We assessed the standard distance-based OoD detection: it labels an instance as OoD if its latent vector is farther than a pre-defined threshold from any

class-conditional mean latent vector of learned in-distribution sets, in terms of the Mahalanobis distance [24] (see Section G of SM for details). Table 5 shows the comparison of competing models using standard metrics (AUCROC and AUPRC) for threshold-based OoD detection, demonstrating that the proposed IQVAE excels in the OoD tasks as it accurately learns the data structure.

### 4.3 SIPaKMeD

The SIPaKMeD dataset includes 4,049 single-cell Pap smear images for cervical cancer diagnosis and is split into 5 classes [32] (see Section C of SM for examples). We assume O(2) symmetry for the SIPaKMeD dataset. We resized the original SIPaKMeD images to size $32 \times 32$ for better usability. In accordance with the original data splitting, 3,549 samples were used for training and 500 samples for testing.

**IQVAE-M.** As we did with rotated MNIST, we conducted the same experiment on the SIPaKMeD dataset. Furthermore, we also as-

Table 6: Clustering ARIs and classification test accuracies for SIPaKMeD (repeated five times).

| METHOD | $k$-MEANS | GMM | SVM | RF |
|---|---|---|---|---|
| AE | $33.4_{\pm 4.0}$ | $35.6_{\pm 1.0}$ | $78.6_{\pm 0.5}$ | $76.9_{\pm 1.1}$ |
| VAE | $33.9_{\pm 1.5}$ | $34.0_{\pm 1.7}$ | $79.4_{\pm 1.0}$ | $76.5_{\pm 2.1}$ |
| $\beta$-VAE | $34.4_{\pm 1.5}$ | $33.7_{\pm 1.1}$ | $79.1_{\pm 1.0}$ | $76.6_{\pm 1.3}$ |
| CRVAE | $24.9_{\pm 3.6}$ | $31.9_{\pm 1.3}$ | $77.1_{\pm 0.7}$ | $80.6_{\pm 1.1}$ |
| QAE | $40.9_{\pm 1.9}$ | $41.1_{\pm 2.9}$ | $80.5_{\pm 0.9}$ | $81.6_{\pm 0.7}$ |
| FMVAE | $39.6_{\pm 0.3}$ | $37.9_{\pm 1.1}$ | $80.0_{\pm 0.8}$ | $75.7_{\pm 1.2}$ |
| QVAE | $40.6_{\pm 3.2}$ | $40.6_{\pm 2.5}$ | $82.2_{\pm 0.8}$ | $\mathbf{81.7_{\pm 0.9}}$ |
| IQVAE | $39.3_{\pm 0.4}$ | $44.7_{\pm 2.0}$ | $82.7_{\pm 0.3}$ | $80.9_{\pm 0.7}$ |
| IQVAE-M | $\mathbf{43.0_{\pm 1.1}}$ | $\mathbf{46.5_{\pm 2.9}}$ | $\mathbf{84.3_{\pm 0.7}}$ | $81.6_{\pm 1.8}$ |

sessed the proposed IQVAE-M that incorporates a learnable flexible Riemannian metric as described in (7) and utilizes the latent Mahalanobis distance when computing the radial basis function (RBF) kernel $K : \mathbb{R}^n \times \mathbb{R}^n \to \mathbb{R}$ of SVMs as $K(\mathbf{z}_0, \mathbf{z}_1) = \exp\left(-(\mathbf{z}_1 - \mathbf{z}_0)^T \mathbf{C}_n (\mathbf{z}_1 - \mathbf{z}_0)\right)$. To quantitatively analyze the learned representations, we present the downstream task performance of the competing models on the SIPaKMeD dataset in Table 6 (see Section E of SM for UMAP visualizations). As shown in Table 6, the proposed methods exhibit superior performance compared to other models[4].

## 5  Conclusion and Limitation

We have proposed and demonstrated the effectiveness of IQVAE, a simple yet effective approach that maintains symmetry and geometry in data. However, our work has two main limitations as follows.

**Predefined groups of symmetry transformations.** We assume the group structure of a given dataset is known in advance. This group invariance is represented using the quotient auto-encoding framework with a Riemannian isometry. However, several recent papers have delved into learning an unknown group directly from data. For example, [35] presents a novel neural network that identifies bispectral invariants. [31] employs Lie algebra to find underlying Lie group invariances. [46] tackles the disentanglement challenge with VAEs, aiming to learn the latent space in terms of one-parameter subgroups of Lie groups. Integrating these methods with our IQVAEs learning approach could be a promising direction for future research.

**Euclidean metric tensors.** We focus on cases where the metric tensor of the observation space is Euclidean. If the intrinsic dimensionality of the image manifold is significantly lower than that of the ambient observation space, the Euclidean metric of the observation space can well serve as a Riemannian metric for the intrinsic geometry of the image manifold. This reasoning supports our use of the pullback metric as a metric tensor for the latent space. However, when constructing a latent representation that effectively addresses a specific task, a specialized metric tensor might be more appropriate than the standard Euclidean metric of the ambient space. For instance, when tasks involve capturing subtle variations in a local image patch, the pullback metric derived from the entire observation dimensions may not provide the most efficient geometry. In such cases, a specialized metric structure better suited for capturing these local variations should be considered. In this context, a task-specific metric can be determined using prior knowledge about the tasks [27]. Alternatively, IQVAEs can semi-supervisedly learn this specialized metric with minimal labeled data, leveraging metric learning concepts [3]. This approach holds significant potential for future research, as it allows the model to tailor its representation to the specific requirements of the task, thereby improving overall performance.

---

[4]When using the Euclidean RBF ($\mathbf{C}_n = \mathbf{I}_n$) for SVMs of the IQVAE-M, the accuracy decreases to 81.9; it shows the usefulness of the learned metric and the corresponding Mahalanobis RBF.

## Acknowledgments and Disclosure of Funding

Changwook Jeong sincerely acknowledges the support from Samsung Electronics Co., Ltd (IO221027-03271-01, IO230220-05060-01) and is further supported by the Technology Innovation Program (20020803, RS-2023-00231956) and Korea Institute for Advancement of Technology (KIAT) grant (P0023703), both funded by the MOTIE, Korea (1415180307, 1415187475), the National Research Foundation of Korea (NRF) grant (RS-2023-00257666) funded by the MSIT, Korea, the IC Design Education Center (IDEC), Korea, and the research project funds (1.220125.01, 1.230063.01) and supercomputing center of UNIST.

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
