# Supplementary Material: IQVAEs

## A  Tips for practical computation

**Quotient reconstruction loss.**    For finite (discrete) groups, the infimum of the quotient reconstruction loss (4) is equal to the minimum of that:

$$\mathcal{L}_{\text{QAE}}(\theta, \phi; \mathbf{x}, G) = \min_{g \in G} \|g * \mathbf{x} - \mu_\phi \circ \mu_\theta(\mathbf{x})\|_2^2,$$

which can be easily computed. For infinite (continuous) groups such as Lie groups, one can discretize such groups and directly apply the above or find the minimum via the gradient descent optimization proposed in [3]. In addition, any type of reconstruction loss, e.g., $L^1$ norm or binary cross-entropy can be used instead of the standard $L^2$ norm. We use the $L^2$ norm and binary cross-entropy for colored and gray-scale image datasets, respectively.

**Jacobians.**    Exact computation of Jacobian in (8) might be a cost bottleneck when optimizing the IQVAEs. One can simply use its first-order approximation given by:

$$\mathbf{J}_{\mu_\phi}^t(\mathbf{z}) \approx \frac{1}{\eta}\big[\mu_\phi(\mathbf{z} + \eta \mathbf{e}_t) - \mu_\phi(\mathbf{z})\big],$$

where $\mathbf{J}_{\mu_\phi}^t$ is the $t$-th column vector of the Jacobian and $\eta$ is a small constant.

An important question is how the trade-off between the estimation quality of the Jacobian and the computational cost in the objective affects the performance of IQVAEs. To check this point, we conducted an experiment to evaluate the estimation quality of the Jacobian using the approximation. The result is shown in Figure S1, where we present the R2 scores between the true pullback metric tensors and their approximations as a function of the number of epochs. It demonstrates the approximation can effectively mimic the true values with R2 greater than 0.96. Furthermore, Table S1 compares the performances of IQVAEs trained with true and approximated Jacobians for the rotated MNIST dataset. It consistently shows that the first-order approximation is a good alternative to the true Jacobian.

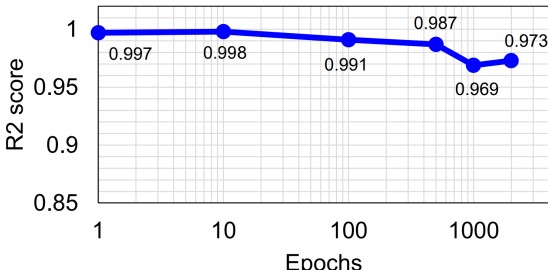

Figure S1: R2 scores between the true and first-order approximated pullback metrics. We used the $2\pi$-rotated MNIST dataset for this experiment. To compute R2, the pullback metric matrices $(N, n, n)$ are reshaped as a vector form $(N, n^2)$ where $N$ is the number of samples and $n$ is the latent dimensionality.

Table S1: Impact of the first-order approximated Jacobian on the final performance of IQVAEs for $2\pi$-rotated MNIST. The column titled "Time [s]" means the training wall-clock time per epoch. We used eight V100 32GB GPUs for this experiment.

| METHOD | $k$-MEANS | GMM | SVM | RF | TIME [S] |
|--------|-----------|------|------|------|---------|
| TRUE   | 71.7      | 70.8 | 92.8 | 92.4 | 316     |
| APPROX.| 70.9      | 72.9 | 92.9 | 92.7 | 3.05    |

## B  Details on $G$-Orbit Pooling

$G$-orbit pooling [2, 3] is a simple yet efficient approach that can achieve $G$-invariance from arbitrary functions. Formally, let $f : \mathbf{x} \mapsto \mathbf{z}$ be an arbitrary function, e.g., neural networks. We denote $\mathbf{z}^i = f^i(\mathbf{x})$ as $i$-th dimensional element of $\mathbf{z}$. We denote $G = \{g_1, ..., g_k, ..., g_K\}$ is a discrete group acting on $\mathbf{x}$ whose $|G| = K$. Then, the $G$-orbit average pooling is given by:

$$f_G^i(\mathbf{x}) = \frac{1}{K} \sum_{k=1}^K f^i(g_k * \mathbf{x}).$$

Similarly, the $G$-orbit max pooling is given by:

$$f_G^i(\mathbf{x}) = \max_{f^i}\{f^i(g_k * \mathbf{x}) | g_k \in G\}.$$

One can easily see such a $f_G$ is a $G$-invariant function, i.e., $f_G(\mathbf{x}) = f_G(g_k * \mathbf{x})$ for all $k = 1, ..., K$. We empirically found that $G$-orbit average pooling and $G$-orbit max pooling exhibit comparable performances for our task. We used the $G$-orbit average pooling for this work.

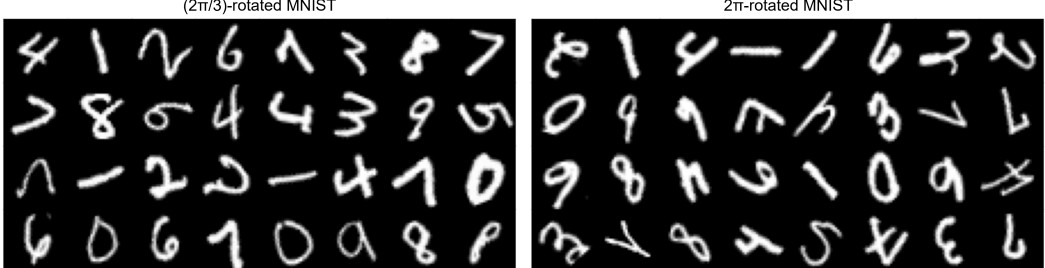

Figure S2: Example images of rotated MNIST datasets.

## C  Example Images of Used Datasets

We visualize some example images of the rotated MNIST, MixedWM38, and SIPakMeD in Figure S2, Figure S3, and Figure S4, respectively. For MixedWM38 and SIPakMeD, each sub-image is titled as its corresponding label.

## D  Implementation Details

We denote fully-connected layers as Dense(input neurons, output neurons). We denote two-dimensional convolutional and transposed convolutional layers as Conv2d(input channel, output channel, kernel size, stride, padding), and ConvT2d (input channel, output channel, kernel size, stride, padding), respectively.

### D.1  Rotated MNIST

Encoders are convolutional neural networks with the following architecture: i) Conv2d(1, 32, 4, 2, same), ii) Conv2d(32, 64, 4, 2, same), iii) Conv2d(64, 64, 4, 2, same), iv) Flatten, v) Dense(1024, 256) with tanh hidden layer activation functions. For AEs and QAEs, the output is Dense(256, 5) layer with linear activation functions. For VAEs, $\beta$-VAEs, CRVAEs, FMVAEs, QVAEs, and IOVAEs, the output is two Dense(256, 5) layers with linear activation functions. For QAEs, QVAEs, and IQVAEs, discrete $G$-orbit pooling is applied to encoders with a rotational group discretized by a step size of $\pi/18$ following [3].

Decoders are transposed convolutional neural networks with the following architecture: i) Dense(5, 256), ii) Dense(256, 1024), iii) Reshape((4, 4, 64)), iv) ConvT2d(64, 64, 4, 2, same), v) ConvT2d(64, 32, 4, 2, same) with tanh hidden layer activation functions. The output is ConvT2d(32, 1, 4, 2, same) with sigmoid activation functions.

All models were trained with a learning rate of $2.5 \times 10^{-4}$ and mini-batch size of 1,024 during 2,000 epochs by using the Adam optimizer [1].

### D.2  MixedWM38

Encoders are convolutional neural networks with the following architecture: i) Conv2d(1, 32, 3, 2, same), ii) Conv2d(32, 32, 3, 2, same), iii) Conv2d(64, 64, 3, 2, valid), iv) Conv2d(64, 64, 3, 2, same), v) Flatten, vi) Dense(576, 256) with tanh hidden layer activation functions. For AEs and QAEs, the output is Dense(256, 5) layer with linear activation functions. For VAEs, $\beta$-VAEs, CRVAEs, FMVAEs, QVAEs, and IOVAEs, the output is two Dense(256, 5) layers with linear activation functions. For QAEs, QVAEs, and IQVAEs, discrete $G$-orbit pooling is applied to encoders with a roto-reflectional group discretized by a step size of $\pi/9$ following [3].

Decoders are transposed convolutional neural networks with the following architecture: i) Dense(5, 256), ii) Dense(256, 576), iii) Reshape((3, 3, 64)), iv) ConvT2d(64, 64, 3, 2, same), v) ConvT2d(64, 32, 3, 2, valid), vi) ConvT2d(32, 32, 3, 2, same) with tanh hidden layer activation functions. The output is ConvT2d(32, 1, 3, 2, same) with sigmoid activation functions.

All models were trained with a learning rate of $2.5 \times 10^{-4}$ and mini-batch size of 1,024 during 3,000 epochs by using the Adam optimizer [1].

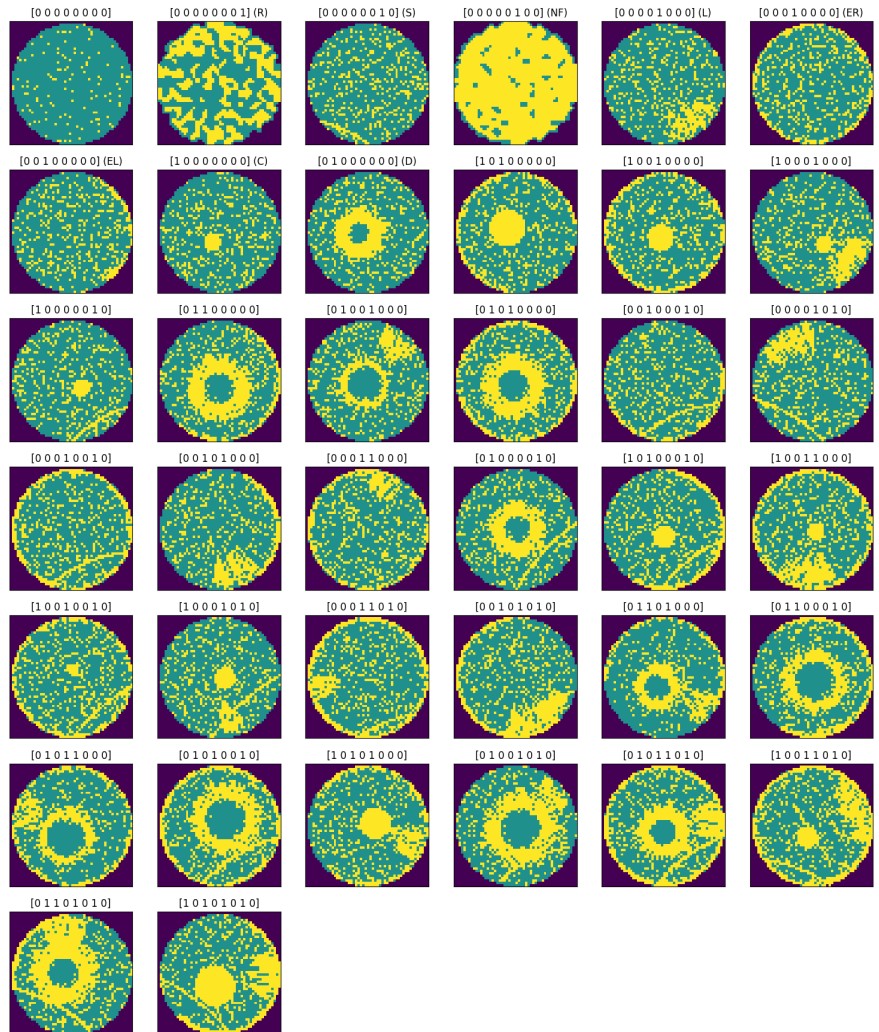

Figure S3: Example images of MixedWM38 dataset [5]. The single-type defects are categorized into 8 patterns: center (C), donut (D), edge-local (EL), edge-ring (ER), local (L), near-full (NF), scratch (S), and random (R). The corresponding labels for each pattern are: center ([1 0 0 0 0 0 0 0]), donut ([0 1 0 0 0 0 0 0]), edge-local ([0 0 1 0 0 0 0 0]), edge-ring ([0 0 0 1 0 0 0 0]), local ([0 0 0 0 1 0 0 0]), near-full ([0 0 0 0 0 1 0 0]), scratch ([0 0 0 0 0 0 1 0]), and random ([0 0 0 0 0 0 0 1]).

## D.3 SIPakMeD

Encoders are convolutional neural networks with the following architecture: i) Conv2d(3, 32, 4, 2, same), ii) Conv2d(32, 64, 4, 2, same), iii) Conv2d(64, 64, 4, 2, same), iv) Flatten, v) Dense(1024, 256) with relu hidden layer activation functions. For AEs and QAEs, the output is Dense(256, 10) layer with linear activation functions. For VAEs, $\beta$-VAEs, CRVAEs, FMVAEs, QVAEs, and IOVAEs, the output is two Dense(256, 10) layers with linear activation functions. For QAEs, QVAEs, and IQVAEs, discrete $G$-orbit pooling is applied to encoders with a roto-reflectional group discretized by a step size of $\pi/9$ following [3].

Decoders are transposed convolutional neural networks with the following architecture: i) Dense(10, 256), ii) Dense(256, 1024), iii) Reshape((4, 4, 64)), iv) ConvT2d(64, 64, 4, 2, same), v) ConvT2d(64, 32, 4, 2, same) with relu hidden layer activation functions. The output is ConvT2d(32, 3, 4, 2, same) with sigmoid activation functions.

All models were trained with a learning rate of $2.5 \times 10^{-4}$ and mini-batch size of 256 during 3,000 epochs by using the Adam optimizer [1].

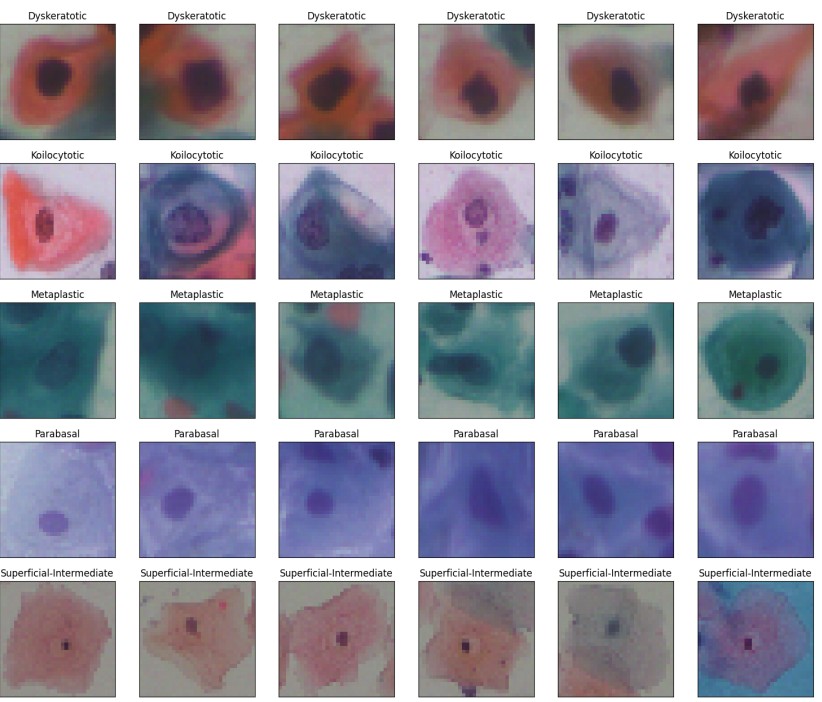

Figure S4: Example images of SIPakMeD dataset [4].

## E  Additional UMAP Visualizations of Latent Representations

We visualize the learned latent representation for the test datasets of $(2\pi/3)$-rotated MNIST, $2\pi$-rotated MNIST, MixedWM38, and SIPakMeD in Figure S5, Figure S6, Figure S7, and Figure S8, respectively.

## F  Additional Interpolation Results

Similar to Figure 5 (a), we provide comparisons of linear interpolation results for different digit pairs across the VAE, QVAE, and IQVAE models in Figure S9 and Figure S10.

## G  Distance-based OoD Detection

Suppose $X = \{\mathbf{x}_i\}_{i=1}^N$ and $Y = \{y_i\}_{i=1}^N$ are respectively the learned in-distribution dataset and corresponding label set with $K$ unique classes. Suppose $\mathbf{z}_i = \mu_\theta(\mathbf{x}_i)$ is a latent representation of $\mathbf{x}_i$ based on the trained encoder $\mu_\theta : \mathbb{R}^d \to \mathbb{R}^n$. The distance-based OoD detection models $K$ class-conditional latent Gaussian distributions $\mathcal{N}(\mu_K, \boldsymbol{\Sigma}), k = 1, ..., K$ as follows:

$$\mu_k = \frac{1}{N_k} \sum_{i:y_i=k} \mathbf{z}_i, \boldsymbol{\Sigma} = \frac{1}{N} \sum_{k=1}^K \sum_{i:y_i=k} (\mathbf{z}_i - \mu_k)(\mathbf{z}_i - \mu_k)^\mathrm{T},$$

where $N_k$ is the number of samples belonging to $k$-th class. Then, for a test instance $\acute{\mathbf{x}}$ the class-wise Mahalanobis distance is computed as follows:

$$d_k(\acute{\mathbf{z}}, \mu_k) = \sqrt{(\acute{\mathbf{z}} - \mu_k)^\mathrm{T} \boldsymbol{\Sigma}^{-1} (\acute{\mathbf{z}} - \mu_k)},$$

where $\acute{\mathbf{z}} = \mu_\theta(\mathbf{x}')$. Finally, the confidence score is given by:

$$s(\acute{\mathbf{z}}) = -\min_k d_k(\acute{\mathbf{z}}, \mu_k).$$

The instance $\acute{\mathbf{x}}$ is considered as OoD if $s(\acute{\mathbf{z}}) \leq \delta$ for a pre-defined threshold $\delta$. By varying $\delta$, one can obtain the receiver operating characteristic (ROC) and precision-recall (PR) curves of OoD detectors. AUCROC and AUPRC are the areas underneath the entire ROC and PR curves, respectively.

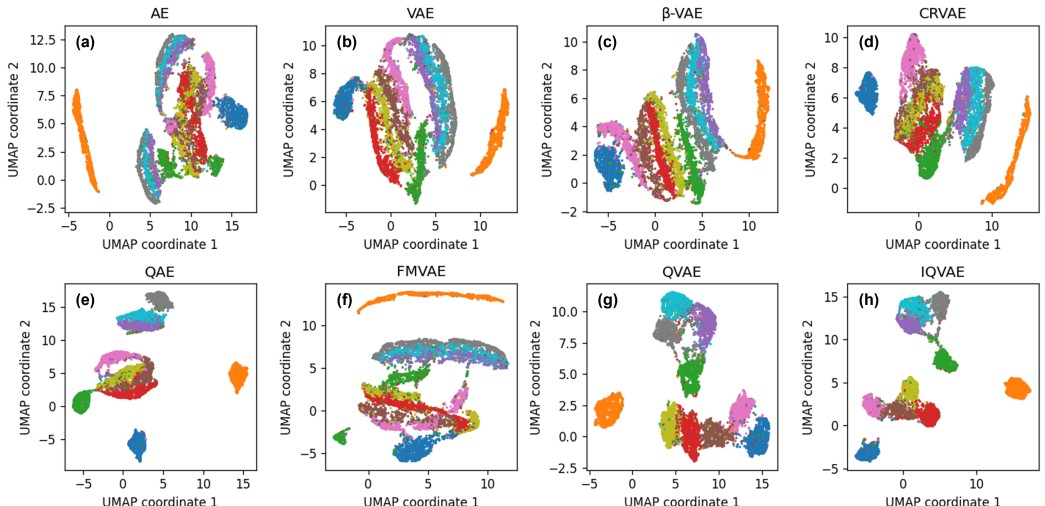

Figure S5: Latent visualization of (a) AE, (b) VAE, (c) $\beta$-VAE, (d) CRVAE, (e) QAE, (f) FMVAE, (g) QVAE and (h) IQVAE for the $(2\pi/3)$-rotated MNIST. The colored clouds represent digits; for example, the blue cloud signifies the digit zero.

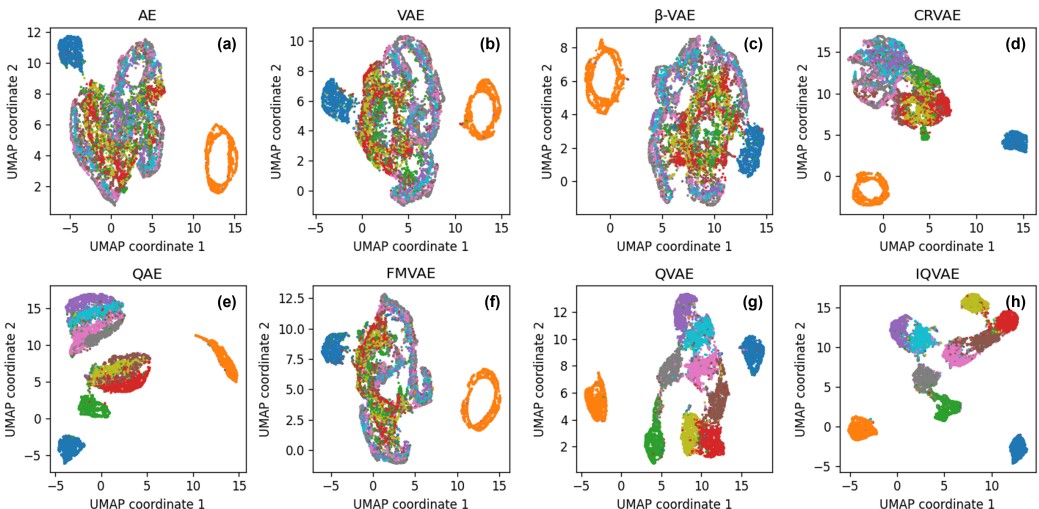

Figure S6: Latent visualization of (a) AE, (b) VAE, (c) $\beta$-VAE, (d) CRVAE, (e) QAE, (f) FMVAE, (g) QVAE and (h) IQVAE for the $2\pi$-rotated MNIST. The colored clouds represent digits; for example, the blue cloud signifies the digit zero.

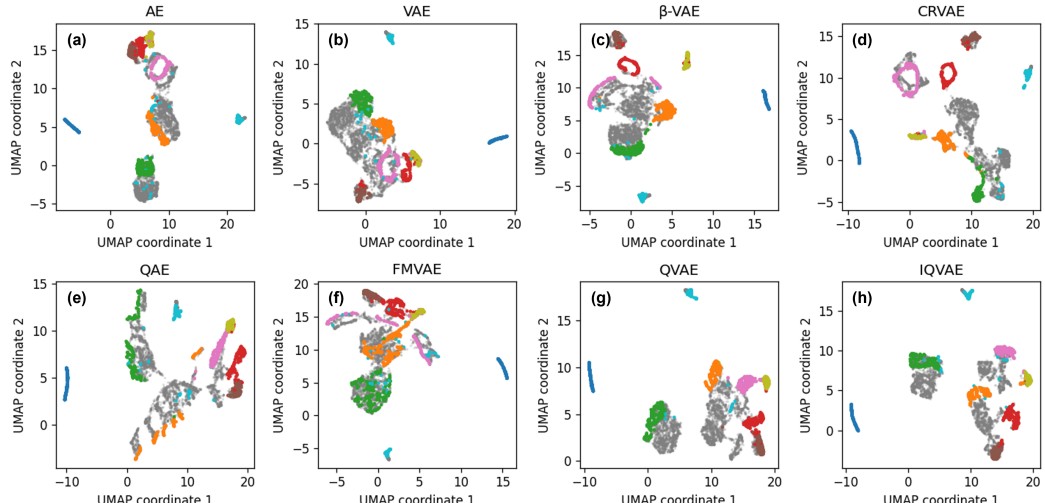

Figure S7: Latent visualization of (a) AE, (b) VAE, (c) $\beta$-VAE, (d) CRVAE, (e) QAE, (f) FMVAE, (g) QVAE and (h) IQVAE for the MixedWM38. The color clouds correspond to defect classes. To enhance clarity, only the points corresponding to single-type defect classes are color-coded.

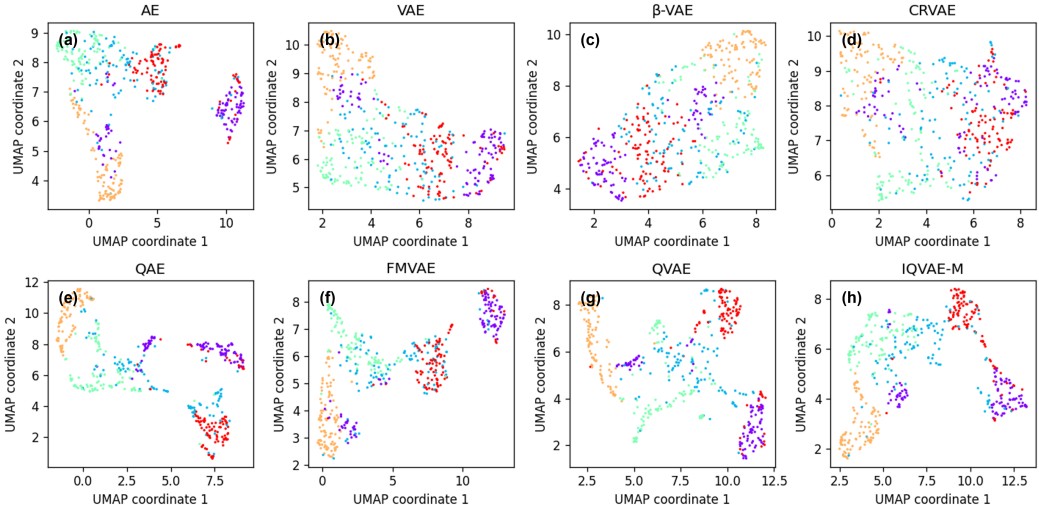

Figure S8: Latent visualization of (a) AE, (b) VAE, (c) $\beta$-VAE, (d) CRVAE, (e) QAE, (f) FMVAE, (g) QVAE and (h) IQVAE for the SIPakMeD. The colored clouds represent specific cell classes.

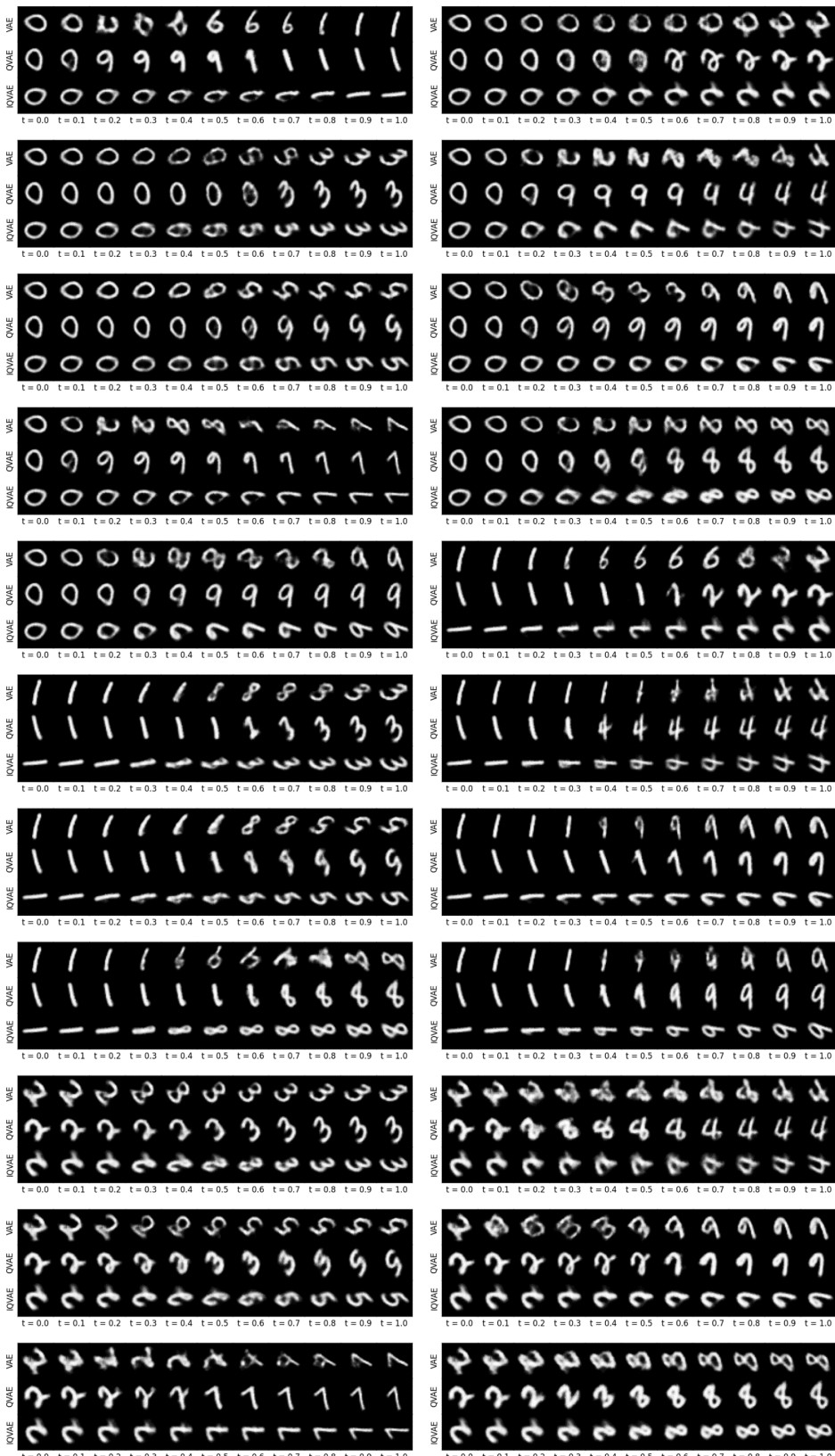

Figure S9: Linear interpolation results for all possible digit combinations (1).

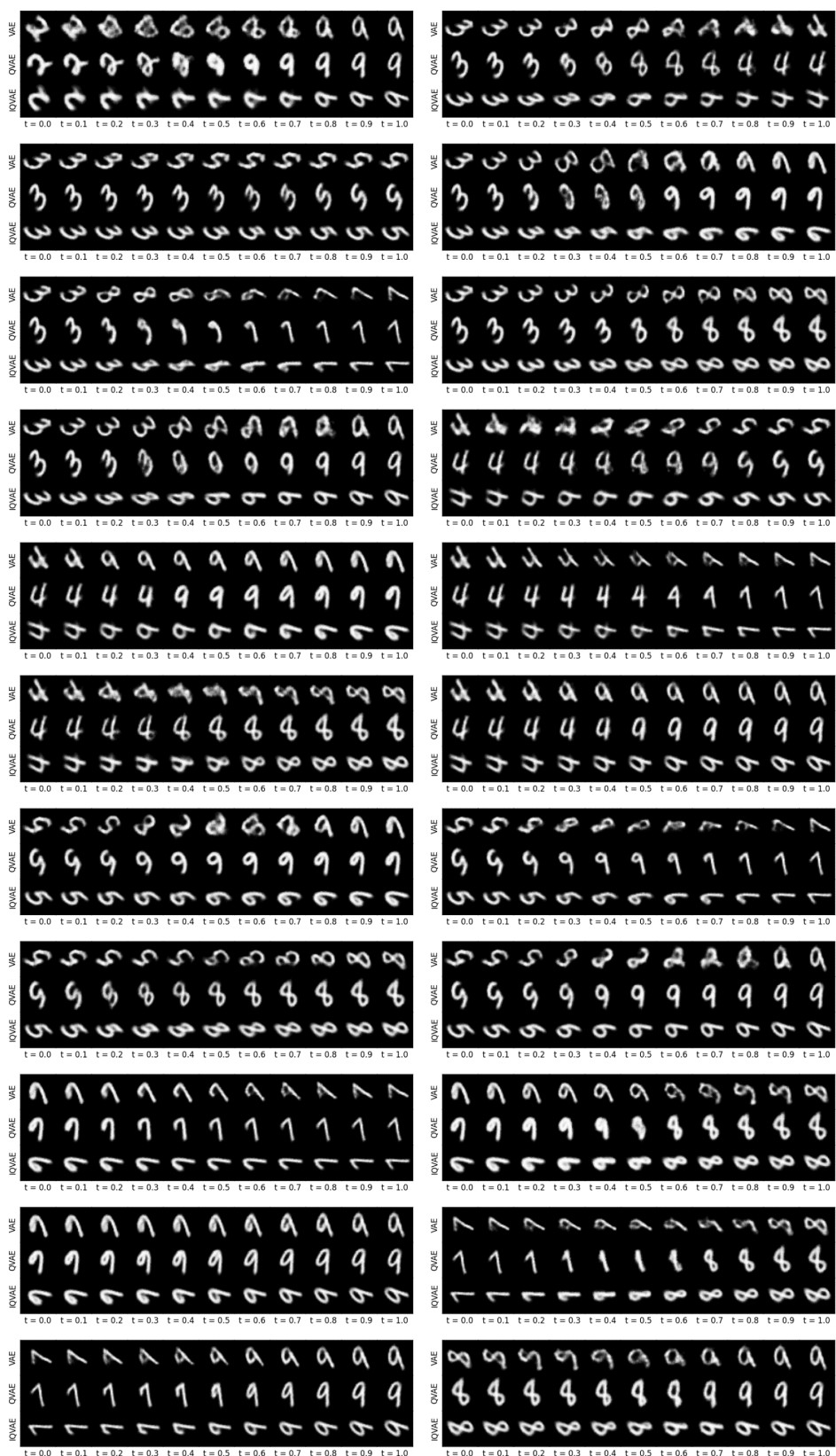

Figure S10: Linear interpolation results for all possible digit combinations (2).