# OpenReview forum: "Isometric Quotient Variational Auto-Encoders for Structure-Preserving Representation Learning"
_NeurIPS.cc/2023/Conference — NeurIPS 2023 poster_

### Official Review · Reviewer_nvnJ · 2023-07-04

**Soundness:** 2 fair
**Presentation:** 4 excellent
**Contribution:** 2 fair
**Rating:** 5
**Confidence:** 3

**Summary:**

This paper considers representation learning and generative models that can learn symmetry and isometric information about the data manifold. Some types of data, especially image data, can have semantic meanings that are invariant under some group of symmetry transformations. Past work has explored models that can learn representations that are invariant or equivariant under such symmetries. This paper proposes to also preserve distance information by learning a representation that is isometrically isomorphic to the data manifold. Two methods based on VAEs are discussed that purport to learn representations that respect symmetry and/or distance structure. These methods are studied in several experiments against notionally similar benchmarks.

**Strengths:**

Thank you to the authors for sharing their interesting research! The provided review of Riemannian geometry is useful, as some readers will not be familiar with these topics.

Originality: Previous work has studied symmetry in representations, and others have studied preserving distance information, but relatively little has been done to respect both aspects simultaneously. The IQVAE training method proposal to directly regularize the Jacobian $J^TJ$ term is interesting.

Quality: The quality of background and method explanation is good. Relevant benchmarks are used in the experiments. Code and adequate reproducibility information is provided.

Clarity: The overall clarity of the paper is good.

Significance: The chosen topic, learning representations and generative models that more accurately reflect the manifold structure of data, is important and it is encouraging to see the authors' efforts in this space.

**Weaknesses:**

Originality: The QVAE loss function proposed in eqn 5 is a very straightforward combination of the VAE and QAE approaches.

Quality: The evidence provided that IQVAE learns isometric embeddings is weak - most of the experiments rely on evaluation metrics on downstream tasks, and use higher performance as evidence that the method is working, but such results do not sufficiently address the claim that isometric embeddings are learned. The only evidence that directly targets distance information seems to be Figure 5b.

Clarity: Because some advanced mathematics is used, some readers may find the methods difficult to follow, but the authors have tried to make that section self-contained with relevant definitions. See questions below for clarifications on some experimental results.

Significance: Wide applicability of the approach is limited by the need to know the symmetry group of the data manifold a priori, and similarly for the metric.

**Questions:**

1. When computing Jacobians for IQVAE, is the approximation in Appendix A used, or are they computed exactly?

2. In Figure 3 TSNE visualizations are shown, and the authors state that they show more "meaningful representations" for IQVAE. TSNE visalizations are inherently qualitative and their 2d nature means they do not capture much information from the relatively higher dimensional latent representations. Can the authors justify why they think TSNE shows that IQVAE learns more meaningful representations, and the mechanism by which TSNE could identify the isometrically isomorphic embeddings that IQVAE purportedly learns?

3. In Figure 4 IQVAE is shown to have aligned reconstructions, but this would also be expected of other approaches that learn group invariances, including several of the paper's benchmarks. The difference of IQVAE to other approaches is in its distance preservation, which is not demonstrated in Figure 4. Can the authors give their rationale for showing only IQVAE here, or provide comparisons to the benchmarks?

4. In Figure 5 linear interpolations are shown. Can the authors explain why they believe the symmetry or distance-aware representations from QVAE and IQVAE should lead to smooth interpolations, and why smooth interpolations are considered a good thing? The statement "The IQVAE shows the smoothest interpolation ... owing to its isometry regularization." is not sufficiently explained by the theory.

5. In the tables, what do the plus-minus numbers represent (std dev, std err, something else)?

6. In the tables of results authors apply bolding on the best results inconsistently. In particular, many numbers are not bolded despite them being within statistical uncertainty of the best results. For instance in Table 1 left, column SVM, all of CRVAE, QAE, QVAE, and IQVAE are within error of the best result, but only QAE and IQVAE are bolded which seems misleading. Can the authors explain their decisions or correct this throughout the paper?

Minor:

In L16, there is more recent research that the authors may consider referencing, which provides more evidence that natural image datasets agree with the manifold hypothesis:

[A] Pope et al. 'The Intrinsic Dimension of Images and Its Impact on Learning' ICLR 2021

[B] Brown et al. 'Verifying the Union of Manifolds Hypothesis for Image Data' ICLR 2023

In Eqn 6, does the F indicate Frobenius norm? This notation is not described.

In line 243, can the authors provide more explanation on what the "mix-up vicinal distribution" is and why it is related to isometries?

There are some grammatical errors that do not detract from the clarity of the work (non-exhaustive list):

L8 rather -> rather than

L53 rather than them of -> maybe "rather than those of"

**Limitations:**

Some limitations are discussed, but only briefly, and no real indication of how to surmount them is proposed. Broader impacts of the work are not discussed.

----
Summary of Discussion: I have read all reviews and rebuttals for this submission.

To summarize the discussion, the authors have provided a considerable amount of additional evidence supporting their method in their rebuttals, especially around my main concern about IQVAEs learning isometric embeddings. I agreed with many of the points in Reviewer AAhr's review, especially their W1, and point that the "concept and the results do not lead to a strong take-home message" in the submitted version of the paper. The biggest outstanding concern of mine is then around novelty, see Weakness: Originality in my review.

I believe that the authors will be able to improve their paper for a final submission, and make it more clear what the roles of symmetry, isometry, and probability are both in their methods but also experimental results.

---

> ### Author Rebuttal · Authors · 2023-08-09
>
> We sincerely appreciate your time and efforts in reviewing our paper, and the constructive comments. Below are our responses to your questions, along with the global response.
>
> ***
>
> **Q1. QVAE loss.**
>
> QVAEs, a probabilistic auto-encoder that can learn group invariance, might be seemed to be a simple extension of QAEs. However, it is essential to highlight their importance extends further as they serve as a bridge to reach IQVAEs, a novel approach that combines both symmetry and geometry in a unified manner. While it is a principled direction, as you pointed out, previous works have not paid sufficient attention to this integration.
>
> ***
>
> **Q2. Evidence for isometry.**
>
> We agree with your comment that there is no direct evidence that IQVAEs learn the isometric embedding, except for a specific case shown in Figure 5 which might not be entirely general. Therefore, we conducted an additional experiment that can provide stronger evidence that IQVAEs indeed learn isometry. Please refer to our response to Q2 of the global rebuttal for details.
>
> ***
>
> **Q3. Limitation on known groups and metrics.**
>
> Please refer to our responses to Q3 and Q4 of the global rebuttal.
>
> ***
>
> **Q4. Jacobian approximation.**
>
> We used the first-order approximation in Appendix A when estimating Jacobian. We will state this in the revised manuscript.
>
> An important question is how the trade-off between the estimation quality of the Jacobian and the computational cost in the objective affects the performance of IQVAEs. To check this point, we conducted an experiment to evaluate the estimation quality of the Jacobian using the approximation. The result is shown in Figure D of the attached PDF file in the global response, where we present the R2 scores between the true pullback metric and its approximation as a function of the number of epochs. It demonstrates the approximation can effectively mimic the true metric with R2 greater than ~0.97.
>
> Furthermore, we compared the performance of IQVAEs trained with true and approximated Jacobians for the rotated MNIST dataset, as presented in Table B of the attached PDF. The results show that the first-order approximation is a good alternative to the true Jacobian. We will include these results in the revised manuscript.
>
> ***
>
> **Q5. t-SNE.**
>
> We consider the presence of a clustered label distribution on t-SNE as evidence of a meaningful representation due to two reasons: (1) the label information is invariant under SO(2) transformations, and (2) images with similar appearances are assigned identical labels, while those with dissimilar appearances receive distinct labels. Thus, the learned representation that preserves the symmetry and geometry is expected to exhibit well-clustered label distributions. As shown in the t-SNE plot of Figure 3, the t-SNE of IQVAEs shows a clear clustered structure according to digit labels, while VAEs do not. As a result, we conclude IQVAEs demonstrate more meaningful representations.
>
> Nevertheless, we agree with your comment that t-SNE primarily preserves local distance information and may not capture the global features of the data. It raises the concern that t-SNE does not ensure IQVAEs learn the isometric embedding, as you correctly pointed out. Thus, we provide UMAP [1] visualization in Figure E of the attached PDF file in the global response, along with the quantitative evidence discussed in Q2. UMAP is known to better preserve the global structure of data compared to t-SNE. As shown in Figure E, IQVAE once again exhibits a distinct cluster structure compared to other baselines. We will add this result in the revised manuscript.
>
> [1] McInnes et al. UMAP: Uniform Manifold Approximation and Projection for Dimension Reduction. arXiv 2018.
>
> ***
>
> **Q6. Aligned reconstructions.**
>
> As you pointed out, the aligned reconstruction is one of the advantages, yet not a unique feature of the IQVAEs. We will state it explicitly.
>
> ***
>
> **Q7. Smooth interpolations.**
>
> IQVAEs learn the isometry, which means that the pullback metric tensor $\mathbf{J}^T \mathbf{J}$ in the latent space is regularized to be an identical matrix $\mathbf{I}$ (up to constant $c^2$). It means that their volume measures are constant, as both theoretically ($\sqrt{\det{\mathbf{J}^T \mathbf{J}}} = c \cdot \sqrt{\det{\mathbf{I}}} = c$) and empirically (please refer to Q2) guaranteed. Thus, IQVAEs ensure that the decoding function from the latent space to the (quotient) observation space is volume-preserving. This means that there is no stretching or distortion of the embeddings, avoiding sudden changes in the images as the latent space varies gradually. Consequently, IQVAEs enable smooth interpolations of images between different latent points.
>
> The smooth interpolation is important in our paper, because it serves as qualitative evidence for the volume-preserving property of IQVAEs. In addition, more generally in the context of generative models, the smooth interpolation might be good because it encourages the latent space continuity and semantic meaning preservation [2]. We will add this discussion in our revised manuscript.
>
> [2] Berthelot et al. Understanding and Improving Interpolation in Autoencoders via an Adversarial Regularizer. ICLR 2019.
>
> ***
>
> **Q8. Tables.**
>
> We used standard deviation for all plus-minus numbers. We will state this in the revised manuscript. In addition, we will revise the tables to ensure consistent criteria for bolding.
>
> ***
>
> **Q9. Important references.**
>
> Thank you for bringing to our attention recent references [3, 4]. We will include them in our revised manuscript.
>
> [3] Pope et al. The Intrinsic Dimension of Images and Its Impact on Learning. ICLR 2021.
>
> [4] Brown et al. Verifying the Union of Manifolds Hypothesis for Image Data. ICLR 2023.
>
> ***
>
> **Q10. Mix-up.**
>
> Please refer to our response to Q1 of the global rebuttal.
>
> ***
>
> **Q11. Others.**
>
> We will add the definition of the Frobenious norm $|| \cdot ||_{F}$ and correct typos.

---

> > ### Comment · Reviewer_nvnJ · 2023-08-13
> > **Response to rebuttals**
> >
> > The authors have put a considerable amount of effort into their rebuttals. I have read all of the reviews and rebuttals.
> >
> > Q2: The new Figure C is the strongest evidence presented so far that IQVAE actually learns isometries. Such evidence was missing in the original submission.
> >
> > Q4: Thank you for the response, it is more than I asked for! I think the approximation is reasonable, but didn't see it clearly stated in the paper whether it was used in the experiments or not.
> >
> > Q5: I agree that UMAP is a better choice than TSNE for this application, and the additional visualizations do show that QAE, QVAE, and IQVAE learn better structured embeddings than other approaches for this data with rotational symmetry. There is also notable consistency between the methods - each of these three have UMAP embeddings where 4 is close to 9, 9 is close to 6, 8 is close to 3, and 3 is close to 5, all of which I think are intuitive.
> >
> > Q7: This explanation is helpful, thank you. I still believe that the example is not showing that IQVAEs have an isometry preserving property. The QVAE is doing just as well on these smooth interpolations, but it is not encouraged to learn isometries. I note that the digits 1 and 7 are probably the most similar to begin with out of the MNIST digits. The authors could explore different combinations (I realize that no more images can be shared in this review process).

---

> > > ### Author Response · Authors · 2023-08-14
> > > **Thank you for the response**
> > >
> > > Thank you once again for your prompt feedback. We are delighted that our previous response effectively addressed your inquiries concerning the learned isometry (**Q2**) and Jacobian approximation (**Q4**). Your perceptive recognition of the value of UMAP visualizations and the established distance relationships (**Q5**) is also deeply appreciated.
> > >
> > > Regarding **Q7**, we would like to respectfully reiterate that the recently conducted experiment, as illustrated in Figure C of the attached PDF file, provides compelling evidence in support of the claim that IQVAEs learn the isometry, as you acknowledged in your feedback to Q2. Particularly noteworthy is Figure C (b), which showcases that IQVAEs consistently maintain the Riemannian volume measure, in stark contrast to the significantly varied measures observed in the instances of VAEs and QVAEs.
> > >
> > > In addition, we believe that the examples of Figure 5 in the manuscript demonstrate to some extent the volume-preservation characteristics of IQVAEs. In Figure 5 (a), the QVAE maintains the image of '1' until time step $t = 6$, then makes an abrupt transition, generating the image of '7' after $t = 7$. On the other hand, the IQVAE retains the image of '1' until $t = 4$, gradually introducing segments between $t = 5$ and $t = 7$, and then generating the image of '7' after $t = 8$. This aligns with the dynamics of the Riemannian volume measure depicted in Figure 5 (b).
> > >
> > > Nevertheless, we acknowledge that this observation might be subtle, and thus not greatly indicatable. Therefore, as you pointed out, the presented interpolation between similar digits '1' and '7' might not be the most effective example for distinctly illustrating the differences between the models. Therefore, according to your suggestion, we have examined various combinations of two digits, revealing similar interpolation patterns as observed for ‘1’ and ‘7’. For instance, when transiting ‘5’ to ‘6’, we found IQVAEs connect the lower 'o' shape most smoothly. In the revised manuscript's appendix, we will provide visualizations for all conceivable interpolations between pairs of digits ($_{10} C_2 = 45 $ cases). Please understand that due to the current limitation on image uploads, we are unable to provide images directly.
> > >
> > > If you have any further questions, please do not hesitate to let us know. We are more than happy to provide further clarifications to all of your inquiries.

---

> > > > ### Comment · Reviewer_nvnJ · 2023-08-20
> > > > **Summary**
> > > >
> > > > To summarize the discussion, the authors have provided a considerable amount of additional evidence supporting their method in their rebuttals, especially around my main concern about IQVAEs learning **isometric** embeddings. I agreed with many of the points in Reviewer AAhr's review, especially their W1, and point that the "concept and the results do not lead to a strong take-home message" in the submitted version of the paper. The biggest outstanding concern of mine is then around novelty, see **Weakness: Originality** in my review.
> > > >
> > > > I believe that the authors will be able to improve their paper for a final submission, and make it more clear what the roles of symmetry, isometry, and probability are both in their methods but also experimental results. I also hope to see Reviewer AAhr respond to the rebuttal.
> > > >
> > > > I have raised my score.

---

> > > > > ### Author Response · Authors · 2023-08-21
> > > > > **Thank you once again for your feedback**
> > > > >
> > > > > We greatly appreciate the time and effort you dedicated to reviewing our paper, your thoughtful feedbacks and subsequent review update. We are particularly delighted to note that our rebuttal adequately resolved your inquiries including the isometric embedding.
> > > > >
> > > > > Regarding your remaining concern about originality, we would like to highlight that the simultaneous consideration of both *geometry* and *symmetry* received limited attention within the existing research landscape, as outlined in Sections 1 and 2 of the manuscript as well as your initial review. Our approach fills this gap by introducing the *Riemannian isometry of group-invariant representations to quotient spaces*. This mathematical concept is effectively realized using the proposed IQVAE framework. Despite the individual components of the proposed IQVAE being established techniques (e.g., QAE and FMVAE), we believe this contribution is still principled and novel as indicated in the NeurIPS 2023 reviewer guidelines [1].
> > > > >
> > > > > The proposed IQVAE yields a more structured representation, which has practical utility for downstream tasks such as clustering. We have validated this claim by comparing the effectiveness of QAE (which is symmetry-aware only), FMVAE/IRVAE (which is geometry-aware only), and the proposed IQVAE. We believe it is a substantial step toward addressing more challenging data equipped with complicated geometry and group symmetry.
> > > > >
> > > > > ***
> > > > >
> > > > > [1] “Originality: Are the tasks or methods new? *Is the work a novel combination of well-known techniques? (This can be valuable!)*“, excerpted from the NeurIPS 2023 reviewer guidelines.

---

### Official Review · Reviewer_Va6B · 2023-07-05

**Soundness:** 3 good
**Presentation:** 3 good
**Contribution:** 3 good
**Rating:** 7
**Confidence:** 4

**Summary:**

This paper introduces a new VAE with the modified objective; they call it improved Quotient Variational Auto-Encoder or IQVAE. This new structure-preserving VAE incorporates an additional regularization term, the Riemannian isometry loss, to optimize the decoder. This new term enables the decoder to preserve Riemannian isometry while mapping latent vectors into images. Working in an unsupervised manner, IQVAE offers enhanced flexibility for dealing with real-world image tasks such as classification and clustering. The effectiveness of the approach is demonstrated through experiments on various datasets and tasks, where some of them show improved results.

**Strengths:**


-- The intuition of the proposed method is well justified and holds promise. Unlike previous VAEs, it emphasizes preserving the structure between the latent space and image space, which has not been well studied in the previous literature.

--  The mathematical concepts underlying Riemannian geometry are explained.

--  Experiments are conducted across multiple datasets and tasks. When compared with six different baselines, some results show promising initial improvements.

**Weaknesses:**


--  The evaluation was conducted on the Rotated MNIST dataset, belonging to the rotation group $SO(2)$. I wonder whether this framework would also be effective for the $SO(3)$ group, which represents a more challenging space for generative models, especially in producing smooth interpolations.

--  Figure 5(a) compares interpolation results. However, the differing start and end points across various methods raise questions about the fairness of the comparison.

--  Some of the baseline VAEs used for comparison, such as AE, VAE, $\beta$-VAE, are somewhat outdated, dating back to around 2016. Against recent baselines like CRVAE and QVAE, the proposed framework does not always give significant improvements, as shown in Tables 1 and 6 (with some results even worse than CRVAE or QVAE). Given that classification tasks are typically sensitive to model tuning, a minor increase in performance (1 percent) should not be considered as superior.

-- The authors should consider comparisons with some of the latest baselines on Riemannian VAEs, like \href{https://proceedings.neurips.cc/paper_files/paper/2022/file/7bf1dc45f850b8ae1b5a1dd4f475f8b6-Paper-Conference.pdf}{A Geometric Perspective on Variational Autoencoders}

**Questions:**


-- The use of Riemannian geometry in the latent is fine but it will be better to relate to a larger question: What is the natural Riemannian structure of images associated with certain types of image sets? Say all views of an object, or all illumination images of an object under a single pose. Does this approach help answer such questions?

**Limitations:**


The paper discusses some limitations -- the choice of Euclidean metric in the image space and the knowledge of the symmetry group. Perhaps they can also discuss the issues arising from 3D rotations of the objects (not just the 2D rotations as used in the paper).

---

> ### Author Rebuttal · Authors · 2023-08-09
>
> We sincerely appreciate your time and efforts in reviewing our paper, and the constructive comments. Below are our responses to your questions, along with the global response.
>
> ***
>
> **Q1. SO(3) group beyond SO(2).**
>
> We would like to emphasize that we have conducted experiments with two datasets (MixedWM38 and SIPaKMeD) that exhibit both rotational and reflectional group symmetry in addition to the rotated MNIST which has SO(2) group symmetry. We will make the necessary revisions to our manuscript to clarify these points more explicitly.
>
> We agree with your comment that considering more complicated 3-dimensional group actions such as SO(3) and SE(3) is an important direction relevant to our work. However, we would like to emphasize that our focus in this work is two-fold: (1) to formulate a novel framework that can consider both the symmetry and geometry of the data manifold simultaneously, i.e., structure-preserving approach, and (2) to validate the proposed framework for image dataset which naturally exhibit 2-dimensional group action.
>
> We would like to mention that even for the simple rotated MNIST with SO(2), previous methods do not perform very well. For instance, for the clustering, CRVAEs record ARI of 26.1, QAEs record ARI of 36.9, while the proposed IQVAEs record ARI of 70.9. Please see Table 1 of our manuscript for more details. Our analysis reveals a critical reason for their failure, which stems from their insufficient attention to the complete structure-preserving representation of the data manifold. We also demonstrate that this limitation can be addressed using the proposed framework, and we believe that it is a substantial step toward addressing more challenging data equipped with complicated geometry and group symmetry.
>
> Following your comment, we will revise the manuscript to provide more details on our contribution and limitations regarding 3-dimensional group actions. We appreciate your feedback and thank you for bringing this important point to our attention.
>
> ***
>
> **Q2. Interpolation comparison.**
>
> We would like to mention that in Figure 5 all models interpolate between the same starting and ending samples (the first ‘1’ and ‘7’ samples, respectively). The reason they appear differently is that (1) VAEs do not learn the SO(2)-invariance, and (2) in the case of QVAEs and IQVAEs, they generates quotient samples with SO(2)-invariance, yet up to the canonical frame (i.e., angle). Nevertheless, we agree that this comparison, as you pointed out, is limited to a specific case and might not be entirely fair. Therefore, we conducted an additional experiment that can provide a more general evidence that the proposed IQVAEs indeed learn the isometric embedding and can interpolate between arbitrary samples smoothly (please refer to our response to Q7 of Reviewer nvnJ for detailed explanation between the isometry, volume preservation, and smooth interpolation). Please refer to our response to Q2 of the global rebuttal for the experimental setting and results.
>
> ***
>
> **Q3. Minor increase in classification tasks.**
>
> We would like to respectfully assert that the proposed IQVAE outperforms the baseline QAE, especially in terms of the fully unsupervised clustering tasks. As we mentioned above, the IQVAE outperforms the competitors, including QAE, by a significantly large margin for the fully-rotated MNIST example, as indicated by an ARI of 70.9 for the IQVAE + k-means, representing a margin of +34.0 over that of the QAE + k-means. Please see Table 1 in our manuscript for more details. In addition, our results show that the IQVAEs (or IQVAE-M) consistently exhibit better performances than competitors for real-world examples such as MixedWM38 and SIPaKMeD. For more details, please refer to Tables 1, 4 and 6 of our manuscript.
>
> Regarding the performance gain of IQVAEs for classification tasks, we agree that it may not be as significant as that for clustering tasks, especially where the sample size is sufficiently large. However, when the sample size is limited, the proposed IQVAEs outperform the QAE baseline even for the classification task with larger margin (~ 4%) as shown in Table 2 of our manuscript. This result has been acknowledged as impressive by Reviewer AAhr. We believe this evidence supports the superiority of our proposed IQVAE in the classification task as well.
>
> ***
>
> **Q4. Important reference Chadebec et al.**
>
> The reference you pointed out [1] is definitely relevant to our work, because it considers the Riemannian geometry of the VAE model. More specifically, [1] proposes a geometry-aware sampling technique that leverages the covariance matrix of the learned encoding distribution as a metric tensor for the latent space. By employing this sampling approach, the vanilla VAE demonstrates improved capability in generating more realistic and plausible samples with ease.
>
> Despite its relevance to our work, we would like to mention that comparison with [1] is not straightforward because they do not modify the learning objective or architecture of the vanilla VAE, and rather introduce the covariance-based Riemannian metric and corresponding sampling technique for the vanilla VAE. As a result, without considering the covariance metric tensor explicitly, the learned latent representation of [1] is identical to that of the vanilla VAEs. It should be noted that conventional downstream models such as k-means cannot consider such a specialized metric structure. Consequently, applying [1] directly to downstream tasks would yield same results with the vanilla VAE. Modulating the downstream model to accommodate special metrics can be possible, yet beyond the scope of the goal of this paper. We will discuss this matter in the revised manuscript.
>
> [1] Chadebec et al. A Geometric Perspective on Variational Autoencoders. NeurIPS 2022.
>
> ***
>
> **Q5. Natural Riemannian structure.**
>
> Please refer to our response to Q3 of the global rebuttal.

---

### Official Review · Reviewer_kuEf · 2023-07-06

**Soundness:** 3 good
**Presentation:** 2 fair
**Contribution:** 2 fair
**Rating:** 4
**Confidence:** 4

**Summary:**

The authors proposed isometric quotient VAEs (IQVAEs), to extract the quotient space and learn the Riemannian isometry. Empirical proof-of-concept experiments are conducted to demonstrate the performance of the method.

**Strengths:**

A new Auto-Encoders approach is proposed and its effectiveness is demonstrated to some extent.

**Weaknesses:**

However, some technical details are not clearly described.

**Questions:**

1. The details of the downstream classification tasks are not clear enough. Line 260, ``tasks by using the support vector machine (SVM)
261 [14] and random forest (RF) classifiers". The author needs to provide the parameters so that others can reproduce the results.
2. The authors need to analyze the results of experiments, such as Table 4 and Table 5.
3. The format of references needs to be standardized.

---

> ### Author Rebuttal · Authors · 2023-08-09
>
> We sincerely appreciate your time and efforts in reviewing our paper, and the constructive comments. Below are our responses to your questions.
>
> ***
>
> **Q1. Details of the downstream classification tasks.**
>
> For the downstream classification model, we utilized the default hyper-parameters provided by scikit-learn. We will state this in the revised manuscript.
>
> We further conducted additional evaluations by varying the regularization strength ($C$) for the SVM model and the number of trees ($n_{estimators}$) for the RF model, to check the influence of the hyper-parameters of downstream classifiers. As shown in Figure A of the attached PDF file in the global response, we found (1) the proposed QVAE and IQVAE consistently outperform the other baselines, and (2) classification accuracy remains relatively consistent throughout the hyper-parameter variations, especially for QVAE and IQVAEs. It further supports the efficiency of the proposed framework in terms of the robustness to the hyper-parameter variations. We will add this experimental result with related discussion in the revised manuscript.
>
> ***
>
> **Q2. Analyzing the results of experiments.**
>
> Table 4 and Table 5 in our manuscript respectively summarizes the downstream task and OoD detection performances for MixedWM38. From line 329 to line 364, we analyzed the used dataset, experimental procedure, and evaluation results.
>
> We would like to enhance the analysis of the experimental results according to your suggestion. If there are specific analysis results you would like to see, please do not hesitate to let us know. We value your input and strive to make our work as rigorous and impactful as possible.
>
> ***
>
> **Q3. Format of references.**
>
> Based on your suggestion, we will standardize the format of references more consistently. We will use the format provided by the NeurIPS 2023 example paper.

---

### Official Review · Reviewer_AAhr · 2023-07-08

**Soundness:** 3 good
**Presentation:** 3 good
**Contribution:** 2 fair
**Rating:** 4
**Confidence:** 3

**Summary:**

This paper proposes an autoencoder framework with additional features of (simultaneously) (1.) An invariance/equivariance of the latent space to a chosen group symmetry (2.) a near local isometry of the latent space. The authors build on the Quotient Autoencoder (QAE) framework that accounts for the group symmetry by modifying it into a probabilistic formulation of a variational autoencoder (called QVAE) and then further adding an isometry condition to the latent space by forcing the metric (computed via jacobians) to be close to a desirable form like identity - IQVAE.

Experiments are demonstrated on the image datasets like Rotated MNIST, MixedWM38, and 3SIPaKMeD. Overall, the results are favorable and marginally outperform all conceptual baselines like AE, VAE, QAE, FMVAE, etc.


**Strengths:**

- Overall, the paper has been compiled well, with decent evaluations and visualizations demonstrating the positive aspects of the contributions of the IQVAE.
- In particular, I found Table 2 and Figure 5 to be interesting experiments: showing performance with limited data and a more meaningful interpolation of the proposed IQVAE


**Weaknesses:**

- I do not get the impression that this paper makes any fundamentally new observation/contribution. The central premise is to combine already known components of QAE, VAE, and Iso-VAE [6, 22] etc - leading to the QVAE and IQVAE. More significantly, the results are decent but not to the extent of making a strong impression, thereby justifying the interest in investigating the combination.
- An important reference must be included, compared, and commented on:   Gropp, Amos, Matan Atzmon, and Yaron Lipman. "Isometric autoencoders." arXiv preprint arXiv:2006.09289 (2020).
- I do not see IRVAE [22] being amongst the baseline evaluations. I understand the symmetry aspect is not a part of this paper. Still, it would be nice to clearly highlight the relative importance of symmetry, isometry, and probabilistic (i.e. like a VAE) components of the autoencoder approach - either implicitly or explicitly using an ablation study. Right now, despite the comparisons in Tables 1, 4, and 5 the message is not clear.



**Questions:**

- More fundamentally, what is the difference between the isometry assumption applied to the decoder (like this paper) in comparison to the encoder?

- What is the importance of the shuffle and augment in Algorithm 1?

**Limitations:**

I do not see any direct societal limitations. See Weaknesses for technical limitations.

Overall, I am voting for a borderline reject at this point. The paper is compiled ok, and reasonably motivated - it is interesting to combine symmetry and isometry into an autoencoder framework. However, both the concept and the results do not lead to a strong take-home message, especially in the midst of the many AE frameworks proposed in prior work.

---

> ### Author Rebuttal · Authors · 2023-08-09
>
> We sincerely appreciate your time and efforts in reviewing our paper, and the constructive comments. Below are our responses to your questions, along with the global response.
>
> ***
>
> **Q1. Combination of already known components.**
>
> We would like to emphasize that our approach of combining both symmetry and geometry in a unified manner is a natural yet principled direction. Previous works, however, have not paid sufficient attention to this integration. As we explained in Section 2 of the manuscript, symmetry-aware works (such as QAEs or CRVAEs) have not adequately considered the underlying geometric distance structure, while geometry-aware works (such as FMVAEs or IRVAEs) have not incorporated the underlying group symmetry. As a result, previous methods do not perform well even for simple rotated MNIST with SO(2) group symmetry. For instance, for the clustering, QAEs record ARI of 36.9, FMVAEs record ARI of 10.1, and the proposed IQVAEs record ARI of 70.9 (please see Table 1 in the manuscript for more details).
>
> We identify a critical reason for the failure of previous methods, which stems from their insufficient attention to the complete structure-preserving representation of the data manifold. We believe that our proposed framework addresses this issue and offers a substantial step toward addressing more challenging data equipped with complicated geometry and group symmetry.
>
> ***
>
> **Q2. Important reference Gropp et al.**
>
> Thank you for bringing the important reference [1] to our attention. We fully agree that the reference is relevant to our work, as it deals with the isometry regularization for auto-encoders to preserve a meaningful distance relationship between samples. We appreciate your suggestions, and we will include this reference in Section 2.2 of the revised manuscript.
>
> The core concept of [1] is similar to that of FMVAEs and IRVAEs, and it still has the limitation of not considering the underlying group symmetry of data. Nonetheless, [1] exhibits the following two intriguing features:
>
> (1) Unlike FMVAEs and IRVAEs, it takes into account the (pseudo) isometry of the encoder.
>
> (2) It proposes a stochastic objective that encourages Jacobian orthogonality (isometry) more cost-effectively.
>
> Regarding case (1), we will provide a related discussion in the response of Q4. For case (2), it can be applied from the perspective of enhancing the learning efficiency of IQVAE. We will discuss this matter in the revised manuscript.
>
> [1] Gropp et al. Isometric Autoencoders. arXiv 2020.
>
> ***
>
> **Q3. IRVAE baseline.**
>
> IRVAEs, as briefly mentioned in Q2, share a limitation with FMVAEs, as they do not take into account the underlying group symmetry explicitly. Consequently, this may result in similar performance outcomes for both models. However, we fully agree with your comment that the comparison with IRVAEs are important for the completeness of the paper as pointed out. Therefore, we evaluated IRVAEs on the fully-rotated MNIST dataset as a preliminary example. As shown in the Table A of the attached PDF file in the global response, it is evident that IRVAEs show comparable performance with FMVAEs. We will include the evaluation results of IRVAEs, including the above experiment, in the revised manuscript.
>
> ***
>
> **Q4. Isometry assumption applied to the decoder.**
>
> The isometry assumption applied to the decoder is particularly important for IQVAEs, because it is desired to match the pullback metric of the latent space $\mathcal{Z}$ with the (Euclidean) metric of the quotient space $\mathcal{X}/G$, rather than that of the observation space $\mathcal{X}$ itself, to prevent from estimating a non-zero distance between two identical images with different poses. It means that the smooth map between $\mathcal{Z}$ and $\mathcal{X}/G$ should be isometry. Note that the encoder and decoder parts of IQVAEs are maps $\mathcal{X} \to \mathcal{Z}$ and $\mathcal{Z} \to \mathcal{X}/G$, respectively. As a result, regularizing the decoder rather than encoder is natural for IQVAEs. We will add this discussion in the revised manuscript.
>
> ***
>
> **Q5. Importance of the shuffle and augment.**
>
> Please refer to our response to Q1 of the global rebuttal.

---

### Author Rebuttal · Authors · 2023-08-09

Dear Reviewers,

We would like to express our heartfelt appreciation for taking the time to review our paper. Your insightful comments and suggestions have been tremendously valuable to us. Below are our responses to some of the frequently asked questions you've shared.

***

**Q1. Mix-up.**

We used the mix-up (shuffle and augment) technique in order to regularize the entire space of interest (i.e., the whole latent space $\mathcal{Z}$) to be isometrically isomorphic to the quotient space. By augmenting the latent space with the mix-up, one can smoothly fill the latent space where data is missing. It can be particularly important when the amount of available training samples is limited.

To support this claim, we conducted an additional experiment that evaluates the effect of the mix-up approach as shown in Figure B of the attached PDF file in the global response. We trained IQVAEs with and without the mix-up augmentation, and computed condition numbers of the pullback metric tensors ($\lambda_{max} / \lambda_{min}$, where $\lambda_{max}$ and $\lambda_{min}$ are respectively the maximum and minimum eigenvalues for the pullback metric $\mathbf{J}^T \mathbf{J}$, when $\mathbf{J}$ is Jacobian of the decoding function) and (mean-normalized) Riemannian volume measures ($\sqrt{\det{\mathbf{J}^T \mathbf{J}}}$) for the test samples. Condition numbers and normalized volume measures near 1.0 with small variances indicate the learned latent representation is isometrically isomorphic. We evaluated both cases with a training sample size of 60,000 and 10,000.

As shown in Figure B, the mix-up approach helps the learned latent space of the IQVAE to be more isometric. In addition, the impact of the mix-up approach becomes more pronounced when the training sample size is reduced. We will add this experimental result with related discussion in the revised manuscript.

***

**Q2. Evidence for isometric embedding.**

To support our claim that IQVAEs learn the isometric embedding, we evaluated the condition numbers of the pullback metric tensors ($\lambda_{max} / \lambda_{min}$, where $\lambda_{max}$ and $\lambda_{min}$ are respectively the maximum and minimum eigenvalues for the pullback metric $\mathbf{J}^T \mathbf{J}$, when $\mathbf{J}$ is Jacobian of the decoding function) and (mean-normalized) Riemannian volume measures ($\sqrt{\det{\mathbf{J}^T \mathbf{J}}}$) for all test samples, across models. Condition numbers and normalized volume measures near 1.0 with small variances indicate the learned latent representation is isometrically isomorphic, thus maintains the volume-preservation.

As shown in Figure C in the attached PDF in the global response, IQVAEs show condition numbers and volume measures near 1.0, which indicate the learned latent representation is desired isometric one. This result can be considered as a generalized version of the analysis reported in Figure 5 (b) in the manuscript. We will add this experimental result in the revised manuscript.

***

**Q3. Natural Riemannian metric and beyond.**

The manifold hypothesis suggests that image data exists as a manifold embedded in a high-dimensional space. When the intrinsic dimensionality of the image manifold is much lower than the observation space, the Euclidean metric of the observation space can serve as a suitable Riemannian metric for the intrinsic geometry of the image manifold. It allows us to capture local distances and angles effectively. Utilizing the induced metric from the ambient Euclidean space facilitates the analysis and modeling of image manifolds within VAEs. It is the reason why we use the pullback metric as a metric tensor for the latent space.

However, when constructing a latent representation for solving a specific task effectively, it might be preferable to use a specialized metric tensor rather than the standard Euclidean metric of the ambient space. For example, when dealing with tasks that require capturing small variations in a local image patch, the pullback metric induced from the entire observation dimensions might not be the most effective geometry. Instead, it is more suitable to consider a specialized metric structure that can efficiently capture such local variations.

In this context, a task-specific metric can be derived using prior knowledge about the tasks. Alternatively, IQVAEs can learn this specialized metric in a semi-supervised manner with very limited labeled data, utilizing the concept of metric learning. This approach holds significant potential for future research, as it enables the model to adapt its representation to the specific requirements of the task, improving overall performance. We will discuss this matter more explicitly in the revised manuscript.

***

**Q4. Limitation on known groups.**

We assume the underlying group structure is known in advance, and represent such a group invariance with a Riemannian isometry by using the auto-encoding framework. On the other hand, there are some recent papers that to learn an unknown group from data. For example, [1] proposes a novel neural network that learns the bispectral invariants. [2] introduces Lie algebra to find underlying Lie group invariances. [3] solves the disentanglement problem with VAEs that can learn the latent space as a form of one-parameter subgroups of Lie groups. Combining these approaches with our proposed learning approach of IQVAEs would be a promising direction for future research. We will elaborate our limitation on known group and the future research direction more explicitly, in the revised manuscript.

[1] Sanborn et al. Bispectral Neural Networks. ICLR 2023.

[2] Moskalev et al. LieGG: Studying Learned Lie Group Generators. NeurIPS 2022.

[3] Zhu et al. Commutative Lie group VAE for disentanglement learning. ICML 2021.

---

### Author Response · Authors · 2023-08-19
**A gentle reminder for reviewers**

Dear Reviewers,

We would like to express our heartfelt appreciation for the dedicated time and thoughtful consideration you have invested in reviewing our paper. Your insightful comments and constructive suggestions have significantly enriched our work.

As the discussion period draws to a close in just a few days, we kindly wish to bring this timeline to your attention. We sincerely hope that we have diligently addressed each of your questions and suggestions through our detailed responses and supporting experiments.

If you find that we have adequately addressed your concerns, we kindly request that you consider updating your review. We understand that this is a voluntary task, and we genuinely appreciate your time and efforts in this regard.

If you have any further questions, please do not hesitate to let us know. We are wholeheartedly committed to providing further elucidations for all your inquiries.

Thank you again for your invaluable feedback, and we look forward to hearing from you soon.

Best regards,

Authors

---

### Decision · Program_Chairs · 2023-09-21

**Decision:**

Accept (poster)

**Comment:**

The paper proposes a new variational auto-encoder training object for capturing data submanifold structures and invariance in the data such as rotations. The performance is tested in three different data sets and shows on-par or improved generation quality.

Initial reviews on the paper are diverging. The authors' response made a notable improvement on the paper and directly addressed criticisms raised on the first version (Reviewer AAhr, nvnJ). Importantly, the primary motivation of the invariance property is tested in the added experiments, such as measuring isometry with condition numbers.

The contribution in the paper is a combined effect: a design on the architecture and losses adapted from existing ones plus experimental results demonstrate some improved performance. To validate a fair contribution, careful revision should be made including articulating design novelties and incorporating the new experimental results.